# Efficient Second-Order Online Kernel Learning with Adaptive Embedding

**Daniele Calandriello**          **Alessandro Lazaric**          **Michal Valko**

SequeL team, INRIA Lille - Nord Europe, France

`{daniele.calandriello, alessandro.lazaric, michal.valko}@inria.fr`

## Abstract

Online kernel learning (OKL) is a flexible framework for prediction problems, since the large approximation space provided by reproducing kernel Hilbert spaces often contains an accurate function for the problem. Nonetheless, optimizing over this space is computationally expensive. Not only first order methods accumulate $\mathcal{O}(\sqrt{T})$ more loss than the optimal function, but the curse of kernelization results in a $\mathcal{O}(t)$ per-step complexity. Second-order methods get closer to the optimum much faster, suffering only $\mathcal{O}(\log T)$ regret, but second-order updates are even more expensive with their $\mathcal{O}(t^2)$ per-step cost. Existing approximate OKL methods reduce this complexity either by limiting the *support vectors* (SV) used by the predictor, or by avoiding the kernelization process altogether using embedding. Nonetheless, as long as the size of the approximation space or the number of SV does not grow over time, an adversarial environment can always exploit the approximation process. In this paper, we propose PROS-N-KONS, a method that combines Nyström sketching to project the input point to a small and accurate embedded space; and to perform efficient second-order updates in this space. The embedded space is continuously updated to guarantee that the embedding remains accurate. We show that the per-step cost only grows with the effective dimension of the problem and not with $T$. Moreover, the second-order updated allows us to achieve the logarithmic regret. We empirically compare our algorithm on recent large-scales benchmarks and show it performs favorably.

## 1 Introduction

*Online learning* (OL) represents a family of efficient and scalable learning algorithms for building a predictive model incrementally from a sequence of $T$ data points. A popular online learning approach [26] is to learn a linear predictor using *gradient descent* (GD) in the input space $\mathbb{R}^d$. Since we can explicitly store and update the $d$ weights of the linear predictor, the total runtime of this algorithm is $\mathcal{O}(Td)$, allowing it to scale to large problems. Unfortunately, it is sometimes the case that no good predictor can be constructed starting from only the linear combination of the input features. For this reason, *online kernel learning* (OKL) [10] first maps the points into a high-dimensional *reproducing kernel Hilbert space* (RKHS) using a non-linear feature map $\varphi$, and then runs GD on the projected points, which is often referred to as functional GD (FGD) [10]. With the kernel approach, each gradient step does not update a fixed set of weights, but instead introduces the feature-mapped point in the predictor as a *support vector* (SV). The resulting kernel-based predictor is flexible and data adaptive, but the number of parameters, and therefore the per-step space and time cost, now scales with $\mathcal{O}(t)$, the number of SVs included after $t$ steps of GD. This *curse of kernelization* results in an $\mathcal{O}(T^2)$ total runtime, and prevents standard OKL methods from scaling to large problems.

Given an RKHS $\mathcal{H}$ containing functions with very small prediction loss, the objective of an OL algorithm is to approach over time the performance of the best predictor in $\mathcal{H}$ and thus minimize the regret, that is the difference in cumulative loss between the OL algorithm and the best predictor *in*

*hindsight*. First-order GD achieve a $\mathcal{O}(\sqrt{T})$ *regret* for any arbitrary sequence of convex losses [10]. However, if we know that the losses are *strongly convex*, setting a more aggressive step-size in first-order GD achieves a smaller $\mathcal{O}(\log T)$ regret [25]. Unfortunately, most common losses, such as the *squared loss*, are not strongly convex when evaluated for a single point $\mathbf{x}_t$. Nonetheless, they posses a certain *directional curvature* [8] that can be exploited by *second-order* GD methods, such as *kernelized online Newton step* (KONS) [2] and kernel-recursive least squares (KRLS) [24], to achieve the $\mathcal{O}(\log T)$ regret without strong convexity along all directions. The drawback of second-order methods is that they have to *store and invert* the $t \times t$ covariance matrix between all SV included in the predictor. This requires $\mathcal{O}(t^2)$ space and time per-step, dwarfing the $\mathcal{O}(t)$ cost of first-order methods and resulting in an even more infeasible $\mathcal{O}(T^3)$ runtime.

**Contributions** In this paper, we introduce PROS-N-KONS, a new OKL method that (1) achieves logarithmic regret for losses with directional curvature using second-order updates, and (2) avoids the curse of kernelization, taking only a fixed per-step time and space cost. To achieve this, we start from KONS, a low-regret *exact* second-order OKL method proposed in [2], but replace the exact feature map $\varphi$ with an approximate $\widetilde{\varphi}$ constructed using a Nyström dictionary approximation. For a dictionary of size $j$, this non-linearly embeds the points in $\mathbb{R}^j$, where we can efficiently perform exact second-order updates in constant $\mathcal{O}(j^2)$ per-step time, and achieve the desired $\mathcal{O}(\log T)$ regret. Combined with an online dictionary learning (KORS [2]) and an adaptive restart strategy, we show that we never get stuck performing GD in an embedded space that is too distant from the true $\mathcal{H}$, but at the same time the size of the embedding $j$ never grows larger than the effective dimension of the problem. While previous methods [13, 11] used *fixed* embeddings, we adaptively construct a small dictionary that scales only with the effective dimension of the data. We then construct an accurate approximation of the covariance matrix, to avoid the variance due to dictionary changes using carefully designed projections.

**Related work** Although first-order OKL methods cannot achieve logarithmic regret, many approximation methods have been proposed to make them scale to large datasets. Approximate methods usually take one of two approaches, either performing approximate gradient updates in the true RKHS (budgeted perceptron [4], projectron [15], forgetron [6]) preventing SV from entering the predictor, or exact gradient updates in an approximate RKHS (Nyström [13], random feature expansion [11]), where the points are embedded in a finite-dimensional space and the curse of kernelization does not apply. Overall, the goal is to never exceed a budget of SVs in order to maintain a fixed per-step update cost. Among budgeted methods, weight degradation [17] can be done in many different ways, such as removal [6] or more expensive projection [15] and merging. Nonetheless, as long as the size of the budget is fixed, the adversary can exploit this to increase the regret of the algorithm, and oblivious inclusion strategies such as uniform sampling [9] fail. Another approach is to replace the exact feature-map $\varphi$ with an approximate feature map $\widetilde{\varphi}$ which allows to explicitly represent the mapped points, and run linear OL on this embedding [13, 21]. When the embedding is oblivious to data, the method is known as random-feature expansion, while a common data-dependent embedding mapping is known as Nyström method [19]. Again, if the embedding is fixed or with a limit in size, the adversary can exploit it. In addition, analyzing a change in embedding during the gradient descent is an open problem, since the underlying RKHS changes with it.

The only approximate second-order method known to achieve logarithmic regret is SKETCHED-KONS. Both SKETCHED-KONS and PROS-N-KONS are based on the exact second-order OL method ONS [8] or its kernelized version KONS [2]. However, SKETCHED-KONS only applies budgeting techniques to the Hessian of the second-order updates and not to the predictor itself, resulting in a $\mathcal{O}(t)$ per-step evaluation time cost. Moreover, the Hessian sketching is performed only through SV removal, resulting in high instability. In this paper, we solve these two issues with PROS-N-KONS by directly approximating KONS using Nyström functional approximation. This results in updates that are closer to SV projection than removal, and that budget both the representation of the Hessian and the predictor.

## 2 Background

**Notation** We borrow the notation from [14] and [2]. We use upper-case bold letters $\mathbf{A}$ for matrices, lower-case bold letters $\mathbf{a}$ for vectors, lower-case letters $a$ for scalars. We denote by $[\mathbf{A}]_{ij}$ and $[\mathbf{a}]_i$ the $(i, j)$ element of a matrix and $i$-th element of a vector respectively. We denote by $\mathbf{I}_T \in \mathbb{R}^{T \times T}$ the identity matrix of dimension $T$ and by $\mathrm{Diag}(\mathbf{a}) \in \mathbb{R}^{T \times T}$ the diagonal matrix with the vector $\mathbf{a} \in \mathbb{R}^T$ on the diagonal. We use $\mathbf{e}_{T,i} \in \mathbb{R}^T$ to denote the indicator vector of dimension $T$ for element $i$.

When the dimension of $\mathbf{I}$ and $\mathbf{e}_i$ is clear from the context, we omit the $T$, and we also indicate the identity operator by $\mathbf{I}$. We use $\mathbf{A} \succeq \mathbf{B}$ to indicate that $\mathbf{A} - \mathbf{B}$ is a positive semi-definite (PSD) matrix. Finally, the set of integers between 1 and $T$ is denoted by $[T] := \{1, \ldots, T\}$.

**Kernels** Given an input space $\mathcal{X}$ and a kernel function $\mathcal{K}(\cdot, \cdot) : \mathcal{X} \times \mathcal{X} \to \mathbb{R}$, we denote the reproducing kernel Hilbert space (RKHS) induced by $\mathcal{K}$ by $\mathcal{H}$, and with $\varphi(\cdot) : \mathcal{X} \to \mathcal{H}$ the associated feature map. Using the feature map, the kernel function can be represented as $\mathcal{K}(\mathbf{x}, \mathbf{x}') = \langle \varphi(\mathbf{x}), \varphi(\mathbf{x}') \rangle_{\mathcal{H}}$, but with a slight abuse of notation we use the simplified notation $\mathcal{K}(\mathbf{x}, \mathbf{x}') = \varphi(\mathbf{x})^\mathsf{T} \varphi(\mathbf{x}')$ in the following. Any function $f \in \mathcal{H}$ can be represented as a (potentially infinite) set of weights $\mathbf{w}$ such that $f_{\mathbf{w}}(\mathbf{x}) = \varphi(\mathbf{x})^\mathsf{T} \mathbf{w}$. Given a set of $t$ points, $\mathcal{D}_t = \{\mathbf{x}_s\}_{s=1}^t$ we denote the feature matrix with $\phi_s$ as its $s$-th column by $\mathbf{\Phi}_t \in \mathbb{R}^{\infty \times t}$.

**Online kernel learning (OKL)** We consider online kernel learning, where an adversary chooses an arbitrary sequence of points $\{\mathbf{x}_t\}_{t=1}^T$ and convex differentiable losses $\{\ell_t\}_{t=1}^T$. The learning protocol is the following. At each round $t \in [T]$ (1) the adversary reveals the new point $\mathbf{x}_t$, (2) the learner chooses a function $f_{\mathbf{w}_t}$ and predicts $f_{\mathbf{w}_t}(\mathbf{x}_t) = \varphi(\mathbf{x}_t)^\mathsf{T} \mathbf{w}_t$, (3) the adversary reveals the loss $\ell_t$, and (4) the learner suffers $\ell_t(\varphi(\mathbf{x}_t)^\mathsf{T} \mathbf{w}_t)$ and observes the associated gradient $\mathbf{g}_t$. We are interested in bounding the cumulative regret between the learner and a fixed function $\mathbf{w}$ defined as $R_T(\mathbf{w}) = \sum_{t=1}^T \ell(\phi_t \mathbf{w}_t) - \ell(\phi_t \mathbf{w})$. Since $\mathcal{H}$ is potentially a very large space, we need to restrict the class of comparators $\mathbf{w}$. As in [14], we consider all functions that guarantee bounded predictions, i.e., $\mathcal{S} = \{\mathbf{w} : \forall t \in [T], |\phi_t^\mathsf{T} \mathbf{w}| \leq C\}$. We make the following assumptions on the losses.

**Assumption 1** (Scalar Lipschitz). *The loss functions $\ell_t$ satisfy $|\ell_t'(z)|$ whenever $|z| \leq C$.*

**Assumption 2** (Curvature). *There exists $\sigma_t \geq \sigma > 0$ such that for all $\mathbf{u}, \mathbf{w} \in \mathcal{S}$ and for all $t \in [T]$,*

$$\ell_t(\phi_t^\mathsf{T} \mathbf{w}) := l_t(\mathbf{w}) \geq l_t(\mathbf{u}) + \nabla l_t(\mathbf{u})^\mathsf{T} (\mathbf{w} - \mathbf{u}) + \frac{\sigma_t}{2} \left( \nabla l_t(\mathbf{u})^\mathsf{T} (\mathbf{w} - \mathbf{u}) \right)^2.$$

This assumption is *weaker than strong convexity* as it only requires the losses to be strongly convex in the direction of the gradient. It is satisfied by squared loss, *squared* hinge loss, and in general, all exp-concave losses [8]. Under this weaker requirement, second-order learning methods [8, 2], obtain the $\mathcal{O}(\log T)$ regret at the cost of a higher computational complexity w.r.t. first-order methods.

**Nyström approximation** A common approach to alleviate the computational cost is to replace the high-dimensional feature map $\varphi$ with a finite-dimensional approximate feature map $\widetilde{\varphi}$. Let $\mathcal{I} = \{\overline{\mathbf{x}}_i\}_{i=1}^j$ be a dictionary of $j$ points from the dataset and $\mathbf{\Phi}_{\mathcal{I}}$ be the associated feature matrix with $\varphi(\overline{\mathbf{x}}_i)$ as columns. We define the embedding $\widetilde{\varphi}(\mathbf{x}) := \mathbf{\Sigma}^{-1} \mathbf{U}^\mathsf{T} \mathbf{\Phi}_{\mathcal{I}}^\mathsf{T} \varphi(\mathbf{x}) \in \mathbb{R}^j$, where $\mathbf{\Phi}_{\mathcal{I}} = \mathbf{V}\mathbf{\Sigma}\mathbf{U}^\mathsf{T}$ is the singular value decomposition of the feature matrix. While in general $\mathbf{\Phi}_{\mathcal{I}}$ is infinite dimensional and cannot be directly decomposed, we exploit the fact that $\mathbf{U}\mathbf{\Sigma}\mathbf{V}^\mathsf{T}\mathbf{V}\mathbf{\Sigma}\mathbf{U}^\mathsf{T} = \mathbf{\Phi}_{\mathcal{I}}^\mathsf{T}\mathbf{\Phi}_{\mathcal{I}} = \mathbf{K}_{\mathcal{I}} = \mathbf{U}\mathbf{\Lambda}\mathbf{U}^\mathsf{T}$ and that $\mathbf{K}_{\mathcal{I}}$ is a (finite-dimensional) PSD matrix. Therefore it is sufficient to compute the eigenvectors $\mathbf{U}$ and eigenvalues $\mathbf{\Lambda}$ of $\mathbf{K}_{\mathcal{I}}$ and take the square root $\mathbf{\Lambda}^{1/2} = \mathbf{\Sigma}$. Note that with this definition we are effectively replacing the kernel $\mathcal{K}$ and $\mathcal{H}$ with an approximate $\mathcal{K}_{\mathcal{I}}$ and $\mathcal{H}_{\mathcal{I}}$, such that $\mathcal{K}_{\mathcal{I}}(\mathbf{x}, \mathbf{x}') = \widetilde{\varphi}(\mathbf{x})^\mathsf{T} \widetilde{\varphi}(\mathbf{x}') = \varphi(\mathbf{x})^\mathsf{T} \mathbf{\Phi}_{\mathcal{I}} \mathbf{U}\mathbf{\Sigma}^{-1}\mathbf{\Sigma}^{-1}\mathbf{U}^\mathsf{T}\mathbf{\Phi}_{\mathcal{I}}^\mathsf{T}\varphi(\mathbf{x}') = \varphi(\mathbf{x})^\mathsf{T}\mathbf{P}_{\mathcal{I}}\varphi(\mathbf{x}')$ where $\mathbf{P}_{\mathcal{I}} = \mathbf{\Phi}_{\mathcal{I}}(\mathbf{\Phi}_{\mathcal{I}}^\mathsf{T}\mathbf{\Phi}_{\mathcal{I}})^{-1}\mathbf{\Phi}_{\mathcal{I}}^\mathsf{T}$ is the projection matrix on the column span of $\mathbf{\Phi}_{\mathcal{I}}$. Since $\widetilde{\varphi}$ returns vectors in $\mathbb{R}^j$, this transformation effectively reduces the computation complexity of kernel operations from $t$ down to the size of the dictionary $j$. The accuracy of $\widetilde{\varphi}$ is directly related to the accuracy of the projection $\mathbf{P}_{\mathcal{I}}$ in approximating the projection $\mathbf{P}_t = \mathbf{\Phi}_t(\mathbf{\Phi}_t^\mathsf{T}\mathbf{\Phi}_t)^{-1}\mathbf{\Phi}_t^\mathsf{T}$, so that for all $s, s' \in [t]$, $\widetilde{\varphi}(\mathbf{x}_s)^\mathsf{T}\widetilde{\varphi}(\mathbf{x}_{s'})$ is close to $\varphi(\mathbf{x}_s)^\mathsf{T}\mathbf{P}_t\varphi(\mathbf{x}_{s'}) = \varphi(\mathbf{x}_s)^\mathsf{T}\varphi(\mathbf{x}_{s'})$.

**Ridge leverage scores** All that is left is to find an efficient algorithm to choose a good dictionary $\mathcal{I}$ to minimize the error $P_{\mathcal{I}} - P_t$. Among dictionary-selection methods, we focus on those that sample points proportionally to their ridge leverage scores (RLSs) [1] because they provide strong reconstruction guarantees. We now define RLS and associated *effective dimension*.

**Definition 1.** *Given a kernel function $\mathcal{K}$, a set of points $\mathcal{D}_t = \{\mathbf{x}_s\}_{s=1}^t$ and a parameter $\gamma > 0$, the $\gamma$-ridge leverage score (RLS) of point $i$ is defined as*

$$\tau_{t,i} = \mathbf{e}_{t,i}\mathbf{K}_t(\mathbf{K}_t + \gamma\mathbf{I}_t)^{-1}\mathbf{e}_{t,i} = \phi_i^\mathsf{T}(\mathbf{\Phi}_t\mathbf{\Phi}_t^\mathsf{T} + \gamma\mathbf{I})^{-1}\phi_i, \tag{1}$$

*and the effective dimension of $\mathcal{D}_t$ as their sum for the each example of $\mathcal{D}_t$,*

$$d_{\textit{eff}}^t(\gamma) = \sum_{i=1}^t \tau_{t,i} = \mathrm{Tr}\left(\mathbf{K}_t(\mathbf{K}_t + \gamma\mathbf{I}_t)^{-1}\right). \tag{2}$$

The RLS of a point measures how orthogonal $\phi_i$ is w.r.t. to the other points in $\mathbf{\Phi}_t$, and therefore how important it is to include it in $\mathcal{I}$ to obtain an accurate projection $\mathbf{P}_\mathcal{I}$. The effective dimension captures the *capacity* of the RKHS $\mathcal{H}$ over the support vectors in $\mathcal{D}_t$. Let $\{\lambda_i\}_i$ be the eigenvalues of $\mathbf{K}_t$, since $d_{\text{eff}}^t(\gamma) = \sum_{s=1}^t \lambda_i/(\lambda_i + \gamma)$, the effective dimension can be seen as the *soft rank* of $\mathbf{K}_t$ where only eigenvalues above $\gamma$ are counted.

To estimate the RLS and construct an accurate $\mathcal{I}$, we leverage KORS [2] (see Alg. 1 in App. A) that extends the online row sampling of Cohen et al. [5] to kernels. Starting from an empty dictionary, at each round, KORS receives a new point $\mathbf{x}_t$, temporarily adds it to the current dictionary $\mathcal{I}_t$ and estimates its associated RLS $\widetilde{\tau}_t$. Then it draws a Bernoulli r.v. proportionally to $\widetilde{\tau}_t$. If the outcome is one, the point is deemed relevant and added to the dictionary, otherwise it is discarded and never added. Note that since points get only evaluated once, and *never dropped*, the size of the dictionary *grows over time* and the RKHS $\mathcal{H}_{\mathcal{I}_t}$ is included in the RKHS $\mathcal{H}_{\mathcal{I}_{t+1}}$, guaranteeing *stability* in the RKHS evolution, unlike alternative methods (e.g., [3]) that construct smaller but often changing dictionaries. We restate the quality of the learned dictionaries and the complexity of the algorithm that we use as a building block.

**Proposition 1** ([2, Thm. 2]). *Given parameters $0 < \varepsilon \leq 1$, $0 < \gamma$, $0 < \delta < 1$, if $\beta \geq 3\log(T/\delta)/\varepsilon^2$ then the dictionary learned by KORS is such that w.p. $1 - \delta$,*

*(1)* *for all rounds $t \in [T]$, we have $\mathbf{0} \preceq \mathbf{\Phi}_t^\mathsf{T}(\mathbf{P}_t - \mathbf{P}_{\mathcal{I}_t})\mathbf{\Phi}_t \preceq +\frac{\varepsilon}{1-\varepsilon}\gamma\mathbf{I}$, and*

*(2)* *the maximum size of the dictionary $J$ is bounded by $\frac{1+\varepsilon}{1-\varepsilon}3\beta d_{\text{eff}}^T(\gamma)\log(2T/\delta)$.*

*The algorithm runs in $\mathcal{O}(d_{\text{eff}}^T(\alpha)^2 \log^4(T))$ space and $\widetilde{\mathcal{O}}(d_{\text{eff}}^T(\alpha)^3)$ time per iteration.*

## 3 The PROS-N-KONS algorithm

We first use a toy OKL example from [2] to illustrate the main challenges for FGD in getting both computational efficiency and optimal regret guarantees. We then propose a different approach which will naturally lead to the definition of PROS-N-KONS.

Consider the case of binary classification with the square loss, where the point presented by the adversary in the sequence is always the same point $\mathbf{x}_{exp}$, but each round with an opposite $\{1, -1\}$ label. Note that the difficulty in this problem arises from the adversarial nature of the labels and it is not due to the dataset itself. The cumulative loss of the comparator $\mathbf{w}$ becomes $(\varphi(\mathbf{x}_{exp})^\mathsf{T}\mathbf{w} - 1)^2 + (\varphi(\mathbf{x}_{exp})^\mathsf{T}\mathbf{w} + 1)^2 + \ldots$ for $T$ steps. Our goal is to achieve $\mathcal{O}(\log T)$ regret w.r.t. the best solution in hindsight, which is easily done by always predicting 0. Intuitively an algorithm will do well when the gradient-step magnitude shrinks as $1/t$. Note that these losses are not strongly convex, thus exact first-order FGD only achieves $\mathcal{O}(\sqrt{T})$ regret and does not guarantee our goal. Exact second-order methods (e.g., KONS) achieve the $\mathcal{O}(\log T)$ regret, but also store $T$ copies of the SV, and have $T^4$ runtime. If we try to improve the runtime using approximate updates and a fixed budget of SV, we lose the $\mathcal{O}(\log T)$ regime, since skipping the insertion of a SV also slows down the reduction in the step-size, both for first-order and second-order methods. If instead we try to compensate the scarcity of SV additions due to the budget with larger updates to the step-size, the adversary can exploit such an unstable algorithm, as is shown in [2] where in order to avoid an unstable solution forces the algorithm to introduce SV with a constant probability. Finally, note that this example can be easily generalized for any algorithm that stores a *fixed* budget of SV, replacing a single $\mathbf{x}_{exp}$ with a set of repeating vectors that exceed the budget. This also defeats oblivious embedding techniques such as random feature expansion with a fixed amount of random features or a fixed dictionary, and simple strategies that update the SV dictionary by insertion and removal.

If we relax the fixed-budget requirement, selection algorithms such as KORS can find an appropriate budget size for the SV dictionary. Indeed, this single sample problem is intrinsically simple: its effective dimension $d_{\text{eff}}^T(\alpha) \simeq 1$ is small, and its induced RKHS $\mathcal{H} = \varphi(\mathbf{x}_{\text{exp}})$ is a singleton. Therefore, following an *adaptive embedding* approach, we can reduce it to a one-dimensional parametric problem and solve it efficiently in this space using exact ONS updates. Alternatively, we can see this approach as constructing an approximate feature map $\widetilde{\varphi}$ that after one step will exactly coincide with the exact feature map $\varphi$, but allows us to run exact KONS updates efficiently replacing $\mathcal{K}$ with $\widetilde{\mathcal{K}}$. Building on this intuition, we propose PROS-N-KONS, a new second-order FGD method that continuously searches for the best embedding space $\mathcal{H}_{\mathcal{I}_t}$ and, at the same time, exploits the small embedding space $\mathcal{H}_{\mathcal{I}_t}$ to efficiently perform exact second-order updates.

We start from an empty dictionary $\mathcal{I}_0$ and a null predictor $\mathbf{w}_0 = \mathbf{0}$. At each round, PROS-N-KONS (Algorithm 1) receives a new point $\mathbf{x}_t$ and invokes KORS to decide whether it should be included in the current dictionary or not. Let $t_j$ with $j \in [J]$ be the *random* step when KORS introduces $\mathbf{x}_{t_j}$ in the dictionary. We analyze PROS-N-KONS as an epoch-based algorithm using these milestones $t_j$. Note that the length $h_j = t_{j+1} - t_j$ and total number of epochs $J$ is random, and is decided in a data-adaptive way by KORS based on the difficulty of the problem. During epoch $j$, we have a fixed dictionary $\mathcal{I}_j$ that induces a feature matrix $\boldsymbol{\Phi}_{\mathcal{I}_j}$ containing samples $\phi_i \in \mathcal{I}_j$, an embedding $\widetilde{\varphi}(\mathbf{x}) : \mathcal{X} \to \mathbb{R}^j = \boldsymbol{\Sigma}_j^{-1} \mathbf{U}_j^{\mathsf{T}} \boldsymbol{\Phi}_j^{\mathsf{T}} \varphi(\mathbf{x})$ based on the singular values $\boldsymbol{\Sigma}_j$ and singular vectors $\mathbf{U}_j$ of $\boldsymbol{\Phi}_j$, with its associated approximate kernel function $\widetilde{\mathcal{K}}$ and induced RKHS $\mathcal{H}_j$. At each round $t_j < t < t_{j+1}$, we perform an *exact* KONS update using the *approximate* map $\widetilde{\varphi}$. This can be computed *explicitly* since $\widetilde{\phi}_t$ is in $\mathbb{R}^j$ and can be easily stored in memory. The update rules are

$$\widetilde{\mathbf{A}}_t = \widetilde{\mathbf{A}}_{t-1} + \frac{\sigma_t}{2} \widetilde{\mathbf{g}}_t \widetilde{\mathbf{g}}_t^{\mathsf{T}}, \ \ \widetilde{\boldsymbol{v}}_t = \widetilde{\boldsymbol{\omega}}_{t-1} - \widetilde{\mathbf{A}}_{t-1}^{-1} \widetilde{\mathbf{g}}_{t-1}, \ \ \widetilde{\boldsymbol{\omega}}_t = \Pi_{\mathcal{S}_t}^{\mathbf{A}_{t-1}}(\boldsymbol{v}_t) = \widetilde{\boldsymbol{v}}_t - \frac{h(\widetilde{\phi}_t^{\mathsf{T}} \widetilde{\boldsymbol{v}}_t)}{\widetilde{\phi}_t^{\mathsf{T}} \widetilde{\mathbf{A}}_{t-1}^{-1} \widetilde{\phi}_t} \widetilde{\mathbf{A}}_{t-1}^{-1} \widetilde{\phi}_t,$$

where the oblique projection $\Pi_{\mathcal{S}_t}^{\mathbf{A}_{t-1}}$ is computed using the closed-form solution from [14]. When $t = t_j$ and a new epoch begins, we perform a *reset step* before taking the first gradient step in the new embedded space. We update the feature-map $\widetilde{\varphi}$, but we reset $\widetilde{\mathbf{A}}_{t_j}$ and $\widetilde{\boldsymbol{\omega}}_{t_j}$ to zero. While this may seem a poor choice, as information learned over time is lost, it leaves intact the dictionary. As long as (a) the dictionary, and therefore the embedded space where we perform our GD, keeps improving and (b) we do not needlessly reset too often, we can count on the fast second-order updates to quickly catch up to the best function in the current $\mathcal{H}_j$. The motivating reason to reset the descent procedure when we switch subspace is to guarantee that our starting point in the descent cannot be influenced by the adversary, and therefore allow us to bound the regret for the overall process (Sect. 4).

**Input:** Feasible parameter $C$, step-sizes $\eta_t$, regularizer $\alpha$
1: Initialize $j = 0, \widetilde{\mathbf{w}}_0 = \mathbf{0}, \widetilde{\mathbf{g}}_0 = \mathbf{0}, \widetilde{\mathbf{P}}_0 = \mathbf{0}, \widetilde{\mathbf{A}}_0 = \alpha \mathbf{I}$,
2: Start a KORS instance with an empty dictionary $\mathcal{I}_0$.
3: **for** $t = \{1, \ldots, T\}$ **do** { Dictionary changed, reset.}
4:     Receive $\mathbf{x}_t$, feed it to KORS.
        Receive $z_t$ (point added to dictionary or not)
5:     **if** $z_{t-1} = 1$ **then**
6:         $j = j + 1$
7:         Build $\mathbf{K}_j$ from $\mathcal{I}_j$ and decompose it in $\mathbf{U}_j \boldsymbol{\Sigma}_j \boldsymbol{\Sigma}_j^{\mathsf{T}} \mathbf{U}_j^{\mathsf{T}}$
8:         Set $\widetilde{\mathbf{A}}_{t-1} = \alpha \mathbf{I} \in \mathbb{R}^{j \times j}$.
9:         $\widetilde{\boldsymbol{\omega}}_t = \mathbf{0} \in \mathbb{R}^j$
10:    **else** {Execute a gradient-descent step.}
11:        Compute map $\phi_t$ and approximate map $\widetilde{\phi}_t = \boldsymbol{\Sigma}_j^{-1} \mathbf{U}_j^{\mathsf{T}} \boldsymbol{\Phi}_j^{\mathsf{T}} \phi_t \in \mathbb{R}^j$.
12:        Compute $\widetilde{\boldsymbol{v}}_t = \widetilde{\boldsymbol{\omega}}_{t-1} - \widetilde{\mathbf{A}}_{t-1}^{-1} \widetilde{\mathbf{g}}_{t-1}$.
13:        Compute $\widetilde{\boldsymbol{\omega}}_t = \widetilde{\boldsymbol{v}}_t - \frac{h(\widetilde{\phi}_t^{\mathsf{T}} \widetilde{\boldsymbol{v}}_t)}{\widetilde{\phi}_t^{\mathsf{T}} \widetilde{\mathbf{A}}_{t-1}^{-1} \widetilde{\phi}_t} \widetilde{\mathbf{A}}_{t-1}^{-1} \widetilde{\phi}_t$
        where $h(z) = \text{sign}(z) \max\{|z| - C, 0\}$
14:   **end if**
15:   Predict $\widetilde{y}_t = \widetilde{\phi}_t^{\mathsf{T}} \widetilde{\boldsymbol{\omega}}_t$.
16:   Observe $\widetilde{\mathbf{g}}_t = \nabla_{\widetilde{\boldsymbol{\omega}}_t} \ell_t(\widetilde{\phi}_t^{\mathsf{T}} \widetilde{\boldsymbol{\omega}}_t) = \ell_t'(\widetilde{y}_t) \widetilde{\phi}_t$.
17:   Update $\widetilde{\mathbf{A}}_t = \widetilde{\mathbf{A}}_{t-1} + \frac{\sigma_t}{2} \widetilde{\mathbf{g}}_t \widetilde{\mathbf{g}}_t^{\mathsf{T}}$.
18: **end for**

Figure 1: PROS-N-KONS

**Computational complexity** PROS-N-KONS's computation complexity is dominated by $\widetilde{\mathbf{A}}_t^{-1}$ inversion required to compute the projection and the gradient update and by the query to KORS, that internally also inverts a $j \times j$ matrix. Therefore, a naïve implementation requires $\mathcal{O}(j^3)$ per-step time and has a space $\mathcal{O}(j^2)$ space complexity necessary to store $\widetilde{\mathbf{A}}_t$. Notice that taking advantage of the fact that KORS only adds SV to the dictionary and never removes them, and that similarly, the $\widetilde{\mathbf{A}}_t$ matrix is constructed using rank-one updates, a careful implementation reduces the per-step cost to $\mathcal{O}(j^2)$. Overall, the total runtime of PROS-N-KONS is then $\mathcal{O}(TJ^2)$, which using the bound on $J$ provided by Prop. 1 and neglecting logarithmic terms reduces to $\widetilde{\mathcal{O}}(T d_{\text{eff}}^T(\gamma)^2)$. Compared to other exact second-order FGD methods, such as KONS or RKLS, PROS-N-KONS dramatically improves the time and space complexity from *polynomial to linear*. Unlike other approximate second-order methods, PROS-N-KONS does not add a new SV at each step. This way it removes $T^2$ from the $\mathcal{O}(T^2 + T d_{\text{eff}}^T(\gamma)^3)$ time complexity of SKETCHED-KONS [2]. Moreover, when $\min_t \tau_{t,t}$ is small, SKETCHED-KONS needs to compensate by adding a constant probability of adding a SV to the dictionary, resulting in a larger runtime complexity, while PROS-N-KONS has *no dependency on the value of the RLS*. Even compared to first-order methods, which incur a larger regret, PROS-N-KONS performs favorably, improving on the $\mathcal{O}(T^2)$ runtime of exact first-order FGD. Compared to other approximate methods, the variant using rank-one updates matches the $\mathcal{O}(J^2)$ per-step cost of the more accurate first-order methods such as the budgeted perceptron [4], projectron [15], Nyström GD [13], while improving on their regret. PROS-N-KONS also closely matches faster but less accurate $\mathcal{O}(J)$ methods such as the forgetron [6] and budgeted GD [23].

## 4 Regret guarantees

In this section, we study the regret performance of PROS-N-KONS.

**Theorem 1** (proof in App. B,). *For any sequence of losses $\ell_t$ satisfying Asm. 2 with Lipschitz constant L, let $\sigma = \min_t \sigma_t$. If $\eta_t \geq \sigma$ for all $t$, $\alpha \leq \sqrt{T}$, $\gamma \leq \alpha$, and predictions are bounded by $C$, then the regret of* PROS-N-KONS *over $T$ steps is bounded w.p. $1 - \delta$ as*

$$R_T(\mathbf{w}) \leq J\left(\alpha\|\mathbf{w}\|^2 + \frac{4}{\sigma}d_{\mathit{eff}}^T\left(\frac{\alpha}{\sigma L^2}\right)\log\left(2\sigma L^2 T/\alpha\right)\right) + \frac{L^2}{\alpha}\left(\frac{T\gamma\varepsilon}{4(1-\varepsilon)}+1\right) + 2JC, \quad (3)$$

*where $J \leq 3\beta d_{\mathit{eff}}^T(\gamma)\log(2T)$ is the number of epochs. If $\gamma = \alpha/T$ the previous bound reduces to*

$$R_T(\mathbf{w}) = \mathcal{O}\left(\alpha\|\mathbf{w}\|^2 d_{\mathit{eff}}^T(\alpha/T)\log(T) + d_{\mathit{eff}}^T(\alpha/T)\,d_{\mathit{eff}}^T(\alpha)\log^2(T)\right). \quad (4)$$

**Remark (bound)** The bound in Eq. 3 is composed of three terms. At each epoch of PROS-N-KONS, an instance of KONS is run on the embedded feature space $\mathcal{H}_j$ obtained by using the dictionary $\mathcal{I}_j$ constructed up to the previous epoch. As a result, we directly use the bound on the regret of KONS (Thm. 1 in [2]) for each of the $J$ epochs, thus leading to the first term in the regret. Since a new epoch is started whenever a new SV is added to the dictionary, the number of epochs $J$ is at most the size of the dictionary returned by KORS up to step $T$, which w.h.p. is $\widetilde{\mathcal{O}}(d_{\mathrm{eff}}^T(\gamma))$, making the first term scale as $\widetilde{\mathcal{O}}(d_{\mathrm{eff}}^T(\gamma)d_{\mathrm{eff}}^T(\alpha))$ overall. Nonetheless, the comparator used in the per-epoch regret of KONS is constrained to the RKHS $\mathcal{H}_j$ induced by the embedding used in epoch $j$. The second term accounts for the difference in performance between the best solutions in the RKHS in epoch $j$ and in the original RKHS $\mathcal{H}$. While this error is directly controlled by KORS through the RLS regularization $\gamma$ and the parameter $\varepsilon$ (hence the factor $\gamma\varepsilon/(1-\varepsilon)$ from Property (1) in Prop. 1), its impact on the regret is amplified by the length of each epoch, thus leading to an overall linear term that needs to be regularized. Finally, the last term summarizes the regret suffered every time a new epoch is started and the default prediction $\widehat{y} = 0$ is returned. Since the values $y_t$ and $\widehat{y}_t$ are constrained in $\mathcal{S}$, this results in a regret of $2JC$.

**Remark (regret comparison)** Tuning the RLS regularization as $\gamma = \alpha/T$ leads to the bound in Eq. 4. While the bound displays an explicit logarithmic dependency on $T$, this comes at the cost of increasing the effective dimension, which now depends on the regularization $\alpha/T$. While in general this could possibly compromise the overall regret, if the sequence of points $\phi_1, \ldots, \phi_T$ induces a kernel matrix with a rapidly decaying spectrum, the resulting regret is still competitive. For instance, if the eigenvalues of $\mathbf{K}_T$ decrease as $\lambda_t = at^{-q}$ with constants $a > 0$ and $q > 1$, then $d_{\mathrm{eff}}^T(\alpha/T) \leq aqT^{1/q}/(q-1)$. This shows that for any $q > 2$ we obtain a regret[1] $o(\sqrt{T}\log^2 T)$ showing that KONS still improves over first-order methods. Furthermore, if the kernel has a low rank or the eigenvalues decrease exponentially, the final regret is poly-logarithmic, thus preserving the full advantage of the second-order approach. Notice that this scenario is always verified when $\mathcal{H} = \mathbb{R}^d$, and is also verified when the adversary draws samples from a stationary distribution and, e.g., the Gaussian kernel [22] (see also [16, 18]). This result is particularly remarkable when compared to SKETCHED-KONS, whose regret scales as $\mathcal{O}(\alpha\|\mathbf{w}\|^2 + d_{\mathrm{eff}}^T(\alpha)(\log T)/\eta)$, where $\eta$ is the fraction of samples which is forced into the dictionary (when $\eta = 1$, we recover the bound for KONS). Even when the effective dimension is small (e.g., exponentially decaying eigenvalues), SKETCHED-KONS requires setting $\eta$ to $T^{-p}$ for a constant $p > 0$ to get a subquadratic space complexity, at the cost of increasing the regret to $\mathcal{O}(T^p \log T)$. On the other hand, PROS-N-KONS achieves a poly-logarithmic regret with linear space complexity up to poly-log factors (i.e., $Td_{\mathrm{eff}}^T(\gamma)^2$), thus greatly improving both the learning and computational performance w.r.t. SKETCHED-KONS. Finally, notice that while $\gamma = \alpha/T$ is the best choice agnostic to the kernel, better bounds can be obtained optimizing Eq. 3 for $\gamma$ depending on $d_{\mathrm{eff}}^T(\gamma)$. For instance, let $\gamma = \alpha/T^s$, then the optimal value of $s$ for $q$-polynomially decaying spectrum is $s = q/(1+q)$, leading to a regret bound $\widetilde{\mathcal{O}}(T^{q/(1+q)})$, which is always $o(\sqrt{T})$ for any $q > 1$.

**Remark (comparison in the Euclidean case)** In the special case $\mathcal{H} = \mathbb{R}^d$, we can make a comparison with existing approximate methods for OL. In particular, the closest algorithm is SKETCHED-ONS by Luo et al. [14]. Unlike PROS-N-KONS, and similarly to SKETCHED-KONS, they take the

approach of directly approximating $\mathbf{A}_t$ in the exact $\mathcal{H} = \mathbb{R}^d$ using *frequent directions* [7] to construct a $k$-rank approximation of $\mathbf{A}_t$ for a *fixed* $k$. The resulting algorithm achieves a regret that is bounded by $k \log T + k \sum_{i=k+1}^{T} \sigma_i^2$, where the sum $\sum_{i=k+1}^{T} \sigma_i^2$ is equal to the sum of all the smallest $d - k$ eigenvalues of the final (exact) matrix $\mathbf{A}_T$. This quantity can vary from 0, when the data lies in a subspace of rank $r \leq k$, to $T\frac{d-k}{d}$ when the sample lie orthogonally and in equal number along all $d$ directions available in $\mathbb{R}^d$. Computationally, the algorithm requires $\mathcal{O}(Tdk)$ time and $\mathcal{O}(dk)$ space. Conversely, PROS-N-KONS automatically adapt its time and space complexity to the effective dimension of the algorithm $d_{\text{eff}}^T(\alpha/T)$ which is smaller than the rank for any $\alpha$. As a consequence, it requires only $\widetilde{\mathcal{O}}(Tr^2)$ time and $\widetilde{\mathcal{O}}(r^2)$ space, achieving a $\mathcal{O}(r^2 \log T)$ regret independently from the spectrum of the covariance matrix. Computationally, all of these complexities are smaller than the ones of SKETCHED-ONS in the regime $r < k$, which is the only one where SKETCHED-ONS can guarantee a sublinear regret, and where the regrets of the two algorithms are close. Overall, while SKETCHED-ONS implicitly relies on the $r < k$ assumption, but continues to operate in a $d$ dimensional space and suffers large regret if $r > k$, PROS-N-KONS will adaptively convert the $d$ dimensional problem into a simpler one with the appropriate rank, fully reaping the computational and regret benefits.

The bound in Thm. 1 can be refined in the specific case of squared loss as follows.

**Theorem 2.** *For any sequence of squared losses $\ell_t = (y_t - \widehat{y}_t)^2$, $L = 4C$ and $\sigma = 1/(8C^2)$, if $\eta_t \geq \sigma$ for all $t$, $\alpha \leq \sqrt{T}$ and $\gamma \leq \alpha$, the regret of* PROS-N-KONS *over $T$ steps is bounded w.p. $1 - \delta$ as*

$$R_T(\mathbf{w}) \leq \sum_{j=1}^{J} \left( \frac{4}{\sigma} d_{\text{eff}}^j \left( \frac{\alpha}{\sigma L^2} \right) \log \left( 2\sigma \frac{L^2}{\alpha} \operatorname{Tr}(\mathbf{K}_j) \right) + \varepsilon' \mathcal{L}_j^* \right) + J \left( L \left( C + \frac{L}{\alpha} \right) + \varepsilon' \alpha \|\mathbf{w}\|_2^2 \right), \quad (5)$$

*where $\varepsilon' = \alpha \left( \alpha - \frac{\gamma\varepsilon}{1-\varepsilon} \right)^{-1} - 1$ and $\mathcal{L}_j^* = \min_{\mathbf{w} \in \mathcal{H}} \left( \sum_{t=t_j}^{t_{j+1}-1} \left( \boldsymbol{\phi}_t^\intercal \mathbf{w} - y_t \right)^2 + \alpha \|\mathbf{w}\|_2^2 \right)$ is the best regularized cumulative loss in $\mathcal{H}$ within epoch $j$.*

Let $\mathcal{L}_T^*$ be the best regularized cumulative loss over all $T$ steps, then $\mathcal{L}_j^* \leq \mathcal{L}_T^*$. Furthermore, we have that $d_{\text{eff}}^j \leq d_{\text{eff}}^T$ and thus regret in Eq. 5 can be (loosely) bounded as

$$R_T(\mathbf{w}) = \mathcal{O} \left( J \left( d_{\text{eff}}^T(\alpha) \log(T) + + \varepsilon' \mathcal{L}_j^* + \varepsilon' \alpha \|\mathbf{w}\|_2^2 \right) \right).$$

The major difference w.r.t. the general bound in Eq. 3 is that we directly relate the regret of PROS-N-KONS to the performance of the best predictor in $\mathcal{H}$ in hindsight, which replaces the linear term $\gamma T/\alpha$. As a result, we can set $\gamma = \alpha$ (for which $\varepsilon' = \varepsilon/(1 - 2\varepsilon)$) and avoid increasing the effective dimension of the problem. Furthermore, since $\mathcal{L}_T^*$ is the regularized loss of the optimal batch solution, we expect it to be small whenever the $\mathcal{H}$ is well designed for the prediction task at hand. For instance, if $\mathcal{L}_T^*$ scales as $\mathcal{O}(\log T)$ for a given regularization $\alpha$ (e.g., in the realizable case $\mathcal{L}_T^*$ is actually just $\alpha \|\mathbf{w}\|$), then the regret of PROS-N-KONS is directly comparable with KONS up to a multiplicative factor depending on the number of epochs $J$ and with a much smaller time and space complexity that adapt to the effective dimension of the problem (see Prop. 1).

## 5 Experiments

We empirically validate PROS-N-KONS on several regression and binary classification problems, showing that it is competitive with state-of-the-art methods. We focused on verifying 1) the advantage of second-order vs. first-order updates, 2) the effectiveness of data-adaptive embedding w.r.t. the oblivious one, and 3) the effective dimension in real datasets. Note that our guarantees hold for more challenging (possibly adversarial) settings than what we test empirically.

**Algorithms** Beside PROS-N-KONS, we introduce two heuristic variants. CON-KONS follows the same update rules as PROS-N-KONS during the descent steps, but at reset steps it does not reset the solution and instead computes $\widetilde{\mathbf{w}}_{t-1} = \boldsymbol{\Phi}_{j-1} \mathbf{U}_{j-1} \boldsymbol{\Sigma}_{j-1}^{-1} \widetilde{\boldsymbol{\omega}}_{t-1}$ starting from $\widetilde{\boldsymbol{\omega}}_{t-1}$ and sets $\widetilde{\boldsymbol{\omega}}_t = \boldsymbol{\Sigma}_j^{-1} \mathbf{U}_j^\intercal \boldsymbol{\Phi}_j^\intercal \widetilde{\mathbf{w}}_{t-1}$. A similar update rule is used to map $\widetilde{\mathbf{A}}_{t-1}$ into the new embedded space without resetting it. B-KONS is a budgeted version of PROS-N-KONS that stops updating the dictionary at a maximum budget $J_{\max}$ and then it continues learning on the last space for the rest of the run. Finally, we also include the best BATCH solution in the final space $\mathcal{H}_J$ returned by KORS as a best-in-hindsight comparator. We also compare to two state-of-the-art embedding-based first-order

| Algorithm | parkinson $n = 5,875$, $d = 20$ | | | cpusmall $n = 8,192$, $d = 12$ | | |
|---|---|---|---|---|---|---|
| | avg. squared loss | #SV | time | avg. squared loss | #SV | time |
| FOGD | **0.04909** $\pm$ 0.00020 | 30 | — | 0.02577 $\pm$ 0.00050 | 30 | — |
| NOGD | **0.04896** $\pm$ 0.00068 | 30 | — | 0.02559 $\pm$ 0.00024 | 30 | — |
| PROS-N-KONS | 0.05798 $\pm$ 0.00136 | 18 | 5.16 | 0.02494 $\pm$ 0.00141 | 20 | 7.28 |
| CON-KONS | 0.05696 $\pm$ 0.00129 | 18 | 5.21 | 0.02269 $\pm$ 0.00164 | 20 | 7.40 |
| B-KONS | 0.05795 $\pm$ 0.00172 | 18 | 5.35 | 0.02496 $\pm$ 0.00177 | 20 | 7.37 |
| BATCH | 0.04535 $\pm$ 0.00002 | — | — | 0.01090 $\pm$ 0.00082 | — | |

| Algorithm | cadata $n = 20,640$, $d = 8$ | | | casp $n = 45,730$, $d = 9$ | | |
|---|---|---|---|---|---|---|
| | avg. squared loss | #SV | time | avg. squared loss | #SV | time |
| FOGD | 0.04097 $\pm$ 0.00015 | 30 | — | 0.08021 $\pm$ 0.00031 | 30 | — |
| NOGD | 0.03983 $\pm$ 0.00018 | 30 | — | 0.07844 $\pm$ 0.00008 | 30 | — |
| PROS-N-KONS | 0.03095 $\pm$ 0.00110 | 20 | 18.59 | **0.06773** $\pm$ 0.00105 | 21 | 40.73 |
| CON-KONS | **0.02850** $\pm$ 0.00174 | 19 | 18.45 | **0.06832** $\pm$ 0.00315 | 20 | 40.91 |
| B-KONS | 0.03095 $\pm$ 0.00118 | 19 | 18.65 | **0.06775** $\pm$ 0.00067 | 21 | 41.13 |
| BATCH | 0.02202 $\pm$ 0.00002 | — | — | 0.06100 $\pm$ 0.00003 | — | — |

| Algorithm | slice $n = 53,500$, $d = 385$ | | | year $n = 463,715$, $d = 90$ | | |
|---|---|---|---|---|---|---|
| | avg. squared loss | #SV | time | avg. squared loss | #SV | time |
| FOGD | **0.00726** $\pm$ 0.00019 | 30 | — | 0.01427 $\pm$ 0.00004 | 30 | — |
| NOGD | 0.02636 $\pm$ 0.00460 | 30 | — | 0.01427 $\pm$ 0.00004 | 30 | — |
| DUAL-SGD | — | — | — | 0.01440 $\pm$ 0.00000 | 100 | — |
| PROS-N-KONS | did not complete | — | — | 0.01450 $\pm$ 0.00014 | 149 | 884.82 |
| CON-KONS | did not complete | — | — | 0.01444 $\pm$ 0.00017 | 147 | 889.42 |
| B-KONS | 0.00913 $\pm$ 0.00045 | 100 | 60 | **0.01302** $\pm$ 0.00006 | 100 | 505.36 |
| BATCH | 0.00212 $\pm$ 0.00001 | — | — | 0.01147 $\pm$ 0.00001 | — | — |

Table 1: Regression datasets

methods from [13]. NOGD selects the first $J$ points and uses them to construct an embedding and then perform exact GD in the embedded space. FOGD uses random feature expansion to construct an embedding, and then runs first-order GD in the embedded space. While oblivious embedding methods are cheaper than data-adaptive Nyström, they are usually less accurate. Finally, DUAL-SGD also performs a random feature expansion embedding, but in the dual space. Given the number #SV of SVs stored in the predictor, and the input dimension $d$ of the dataset's samples, the time complexity of all first-order methods is $\mathcal{O}(Td\#SV)$, while that of PROS-N-KONS and variants is $\mathcal{O}(T(d + \#SV)\#SV)$. When $\#SV \sim d$ (as in our case) the two complexities coincide. The space complexities are also close, with PROS-N-KONS $\mathcal{O}(\#SV^2)$ not much larger than the first order methods' $\mathcal{O}(\#SV)$. We do not run SKETCHED-KONS because the $T^2$ runtime is prohibitive.

**Experimental setup** We replicate the experimental setting in [13] with 9 datasets for regression and 3 datasets for binary classification. We use the same preprocessing as Lu et al. [13]: each feature of the points $\mathbf{x}_t$ is rescaled to fit in $[0, 1]$, for regression the target variable $y_t$ is rescaled in $[0, 1]$, while in binary classification the labels are $\{-1, 1\}$. We also *do not* tune the Gaussian kernel bandwidth, but take the value $\sigma = 8$ used by [13]. For all datasets, we set $\beta = 1$ and $\varepsilon = 0.5$ for all PROS-N-KONS variants and $J_{\max} = 100$ for B-KONS. For each algorithm and dataset, we report average and standard deviation of the losses. The scores for the competitor baselines are reported as provided in the original papers [13, 12]. We only report scores for NOGD, FOGD, and DUAL-SGD, since they have been shown to outperform other baselines such as budgeted perceptron [4], projectron [15], forgetron [6], and budgeted GD [23]. For PROS-N-KONS variant we also report the runtime in seconds, but do not compare with the runtimes reported by [13, 12], as that would imply comparing different implementations. Note that since the complexities $\mathcal{O}(Td\#SV)$ and $\mathcal{O}(T(d + \#SV)\#SV)$ are close, we do not expect large differences. All experiments are run on a single machine with 2 Xeon E5-2630 CPUs for a total of 10 cores, and are averaged over 15 runs.

**Effective dimension and runtime** We use size of the dictionary returned by KORS as a proxy for the effective dimension of the datasets. As expected, larger datasets and datasets with a larger input dimension have a larger effective dimension. Furthermore, $d_{\mathrm{eff}}^T(\gamma)$ increases (sublinearly) when we reduce $\gamma$ from 1 to 0.01 in the *ijcnn1* dataset. More importantly, $d_{\mathrm{eff}}^T(\gamma)$ remains empirically small

| | ijcnn1 $n = 141,691, d = 22$ | | | cod-rna $n = 271,617, d = 8$ | | |
|---|---|---|---|---|---|---|
| $\alpha = 1, \gamma = 1$ | | | | | | |
| Algorithm | accuracy | #SV | time | accuracy | #SV | time |
| FOGD | $9.06 \pm 0.05$ | 400 | — | $10.30 \pm 0.10$ | 400 | — |
| NOGD | $9.55 \pm 0.01$ | 100 | — | $13.80 \pm 2.10$ | 100 | — |
| DUAL-SGD | $\mathbf{8.35} \pm 0.20$ | 100 | — | $\mathbf{4.83} \pm 0.21$ | 100 | — |
| PROS-N-KONS | $9.70 \pm 0.01$ | 100 | 211.91 | $13.95 \pm 1.19$ | 38 | 270.81 |
| CON-KONS | $9.64 \pm 0.01$ | 101 | 215.71 | $18.99 \pm 9.47$ | 38 | 271.85 |
| B-KONS | $9.70 \pm 0.01$ | 98 | 206.53 | $13.99 \pm 1.16$ | 38 | 274.94 |
| BATCH | $8.33 \pm 0.03$ | — | — | $3.781 \pm 0.01$ | — | — |
| $\alpha = 0.01, \gamma = 0.01$ | | | | | | |
| Algorithm | accuracy | #SV | time | accuracy | #SV | time |
| FOGD | $9.06 \pm 0.05$ | 400 | — | $10.30 \pm 0.10$ | 400 | — |
| NOGD | $9.55 \pm 0.01$ | 100 | — | $13.80 \pm 2.10$ | 100 | — |
| DUAL-SGD | $8.35 \pm 0.20$ | 100 | — | $4.83 \pm 0.21$ | 100 | — |
| PROS-N-KONS | $10.73 \pm 0.12$ | 436 | 1003.82 | $4.91 \pm 0.04$ | 111 | 459.28 |
| CON-KONS | $6.23 \pm 0.18$ | 432 | 987.33 | $5.81 \pm 1.96$ | 111 | 458.90 |
| B-KONS | $\mathbf{4.85} \pm 0.08$ | 100 | 147.22 | $\mathbf{4.57} \pm 0.05$ | 100 | 333.57 |
| BATCH | $5.61 \pm 0.01$ | — | — | $3.61 \pm 0.01$ | — | — |

Table 2: Binary classification datasets

even for datasets with hundreds of thousands samples, such as *year*, *ijcnn1* and *cod-rna*. On the other hand, in the *slice* dataset, the effective dimension is too large for PROS-N-KONS to complete and we only provide results for B-KONS. Overall, the proposed algorithm can process hundreds of thousands of points in a matter of minutes and shows that it can practically scale to large datasets.

**Regression** All algorithms are trained and evaluated using the squared loss. Notice that whenever the budget $J_{\max}$ is not exceeded, B-KONS and PROS-N-KONS are the same algorithm and obtain the same result. On regression datasets (Tab. 1) we set $\alpha = 1$ and $\gamma = 1$, which satisfies the requirements of Thm. 2. Note that we *did not* tune $\alpha$ and $\gamma$ for optimal performance, as that would require multiple runs, and violate the online setting. On smaller datasets such as *parkinson* and *cpusmall*, where frequent restarts greatly interfere with the gradient descent, and even a small non-adaptive embedding can capture the geometry of the data, PROS-N-KONS is outperformed by simpler first-order methods. As soon as $T$ reaches the order of tens of thousands (*cadata*, *casp*), second-order updates and data adaptivity becomes relevant and PROS-N-KONS outperform its competitors, both in the number of SVs and in the average loss. In this intermediate regime, CON-KONS outperforms PROS-N-KONS and B-KONS since it is less affected by restarts. Finally, when the number of samples raises to hundreds of thousands, the intrinsic effective dimension of the dataset starts playing a larger role. On *slice*, where the effective dimension is too large to run, B-KONS still outperforms NOGD with a comparable budget of SVs, showing the advantage of second-order updates.

**Binary classification** All algorithms are trained using the hinge loss and they are evaluated using the average online error rate. Results are reported in Tab. 2. While for regression, an arbitrary value of $\gamma = \alpha = 1$ is sufficient to obtain good results, it fails for binary classification. Decreasing the two parameters to 0.01 resulted in a 3-fold increase in the number of SVs included and runtime, but almost a 2-fold decrease in error rate, placing PROS-N-KONS and B-KONS on par or ahead of competitors without the need of any further parameter tuning.

# 6 Conclusions

We presented PROS-N-KONS a novel algorithm for sketched second-order OKL that achieves $\mathcal{O}(d_{\mathrm{eff}}^T \log T)$ regret for losses with directional curvature. Our sketching is data-adaptive and, when the effective dimension of the dataset is constant, it achieves a constant per-step cost, unlike SKETCHED-KONS [2], which was previously proposed for the same setting. We empirically showed that PROS-N-KONS is practical, performing on par or better than state-of-the-art methods on standard benchmarks using small dictionaries on realistic data.

**Acknowledgements** The research presented was supported by French Ministry of Higher Education and Research, Nord-Pas-de-Calais Regional Council, Inria and Univertät Potsdam associated-team north-european project Allocate, and French National Research Agency projects ExTra-Learn (n.ANR-14-CE24-0010-01) and BoB (n.ANR-16-CE23-0003).

## Footnotes

[1]Here we ignore the term $d_{\mathrm{eff}}^T(\alpha)$ which is a constant w.r.t. $T$ for any constant $\alpha$.

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
