[Supplementary Material]

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

# A KORS: Kernel online row sampling

For completeness, we report the KORS algorithm by Calandriello et al. [2] based on online row sampling of Cohen et al. [5] and comment on it.

---

**Algorithm 1** Kernel Online Row Sampling (KORS)

---

**Input:** Regularization $\gamma$, accuracy $\varepsilon$, budget $\beta$
1: Initialize $\mathcal{I}_0 = \emptyset$, $j = 0$
2: **for** $t = \{0, \dots, T-1\}$ **do**
3:      receive $\mathbf{x}_t$, compute $\boldsymbol{\phi}_t$
4:      construct temporary dictionary $\overline{\mathcal{I}} := \mathcal{I}_{t-1} \cup (t, 1)$
5:      compute $\widetilde{p}_t = \min\{\beta \widetilde{\tau}_t, 1\}$ using $\overline{\mathcal{I}}$ and Eq. 6
6:      draw $z_t \sim \mathcal{B}(\widetilde{p}_t)$ and if $z_t = 1$, add $(t, 1/\widetilde{p}_t)$ to $\mathcal{I}_t$ and update $j = j + 1$
7: **end for**

---

At each time step KORS stores and updates a weighted dictionary of points, where we use the notation $\mathcal{I} = \{(i_s, \widetilde{p}_{i_s})\}_{s=1}^j$ to indicate that $\mathcal{I}$ contains point $\boldsymbol{\phi}_i$ in its $s$-th position. The scalar weight $\widetilde{p}_i$ associated with is used when constructing the diagonal reweighing matrix $\mathbf{S}_{\mathcal{I}} = \mathrm{Diag}(\{1/\sqrt{\widetilde{p}_{i_s}}\})$, such that $\boldsymbol{\Phi}_{\mathcal{I}} \mathbf{S}_{\mathcal{I}}$ contains $(1/\sqrt{\widetilde{p}_{i_s}}) \boldsymbol{\phi}_{i_s}$ as its $s$-th column. Starting from an empty dictionary, at each time step KORS receives a new point $\mathbf{x}_t$, temporarily adds it to the current dictionary $\mathcal{I}_t$ and uses Eq. 6,

$$
\begin{aligned}
\widetilde{\tau}_{t,i} &= \frac{(1+\varepsilon)}{\gamma} (\boldsymbol{\phi}_t^{\mathsf{T}} \boldsymbol{\phi}_t - \boldsymbol{\phi}_t^{\mathsf{T}} \boldsymbol{\Phi}_{\overline{\mathcal{I}}} \mathbf{S}_{\overline{\mathcal{I}}} (\mathbf{S}_{\overline{\mathcal{I}}} \boldsymbol{\Phi}_{\overline{\mathcal{I}}}^{\mathsf{T}} \boldsymbol{\Phi}_{\overline{\mathcal{I}}} \mathbf{S}_{\overline{\mathcal{I}}} + \gamma \mathbf{I})^{-1} \mathbf{S}_{\overline{\mathcal{I}}} \boldsymbol{\Phi}_{\overline{\mathcal{I}}}^{\mathsf{T}} \boldsymbol{\phi}_t) \\
&= \frac{1+\varepsilon}{\gamma} \left( k_{t,t} - \mathbf{k}_{\overline{\mathcal{I}},t}^{\mathsf{T}} \mathbf{S}_{\overline{\mathcal{I}}} (\mathbf{S}_{\overline{\mathcal{I}}}^{\mathsf{T}} \mathbf{K}_{\overline{\mathcal{I}}} \mathbf{S}_{\overline{\mathcal{I}}} + \gamma \mathbf{I})^{-1} \mathbf{S}_{\overline{\mathcal{I}}}^{\mathsf{T}} \mathbf{k}_{\overline{\mathcal{I}},t} \right),
\end{aligned} \tag{6}
$$

to compute an estimate RLS $\widetilde{\widetilde{\tau}}_t$ for $\boldsymbol{\phi}_t$. Afterwards, it draws a Bernoulli r.v. $z_t$ proportionally to $\widetilde{\widetilde{\tau}}_t$, if it succeeds ($z_t = 1$) the point is deemed relevant and added to the dictionary, otherwise it is discarded and never added.

# B Proof of Theorem 1: Regret analysis

PROS-N-KONS predicts $\widetilde{y}_t$ in round $t$. We want to bound the cumulative regret of the loss of $\widetilde{y}_t$ with respect to an arbitrary fixed vector $\mathbf{w} \in \mathcal{H}$ on the (mapped) points $\boldsymbol{\phi}_t$. From the definition of $\widetilde{y}_t$ in Algorithm 1, we have

$$
\begin{aligned}
\sum_{t=1}^T \ell_t(\widetilde{y}_t) - \ell_t(\boldsymbol{\phi}_t^{\mathsf{T}} \mathbf{w}) &= \sum_{t=1}^T \ell_t\left(\widetilde{\boldsymbol{\phi}}_t^{\mathsf{T}} \widetilde{\boldsymbol{\omega}}_t\right) - \ell_t(\boldsymbol{\phi}_t^{\mathsf{T}} \mathbf{w}) \\
&= \sum_{t=1}^T \ell_t\left(\boldsymbol{\phi}_t^{\mathsf{T}} \boldsymbol{\Phi}_j \mathbf{U}_j \boldsymbol{\Sigma}_j^{-1} \widetilde{\boldsymbol{\omega}}_t\right) - \ell_t(\boldsymbol{\phi}_t^{\mathsf{T}} \mathbf{w}) \\
&= \sum_{t=1}^T \ell_t(\boldsymbol{\phi}_t^{\mathsf{T}} \widetilde{\mathbf{w}}_t) - \ell_t(\boldsymbol{\phi}_t^{\mathsf{T}} \mathbf{w}),
\end{aligned}
$$

with $\widetilde{\mathbf{w}}_t = \boldsymbol{\Phi}_j \mathbf{U}_j \boldsymbol{\Sigma}_j^{-1} \widetilde{\boldsymbol{\omega}}_t$. From Assumption 2 on the losses, we know that the losses $\ell_t(z)$ satisfy

$$
\ell_t(\boldsymbol{\phi}^{\mathsf{T}} \mathbf{w}) \geq \ell_t(\boldsymbol{\phi}^{\mathsf{T}} \mathbf{u}) + \nabla \ell_t(\boldsymbol{\phi}^{\mathsf{T}} \mathbf{u})^{\mathsf{T}} (\mathbf{w} - \mathbf{u}) + \frac{\sigma_t}{2} \left(\nabla \ell_t(\boldsymbol{\phi}^{\mathsf{T}} \mathbf{u})^{\mathsf{T}} (\mathbf{w} - \mathbf{u})\right)^2
$$

and therefore

$$
\begin{aligned}
\ell_t(\boldsymbol{\phi}^{\mathsf{T}} \mathbf{u}) - \ell_t(\boldsymbol{\phi}^{\mathsf{T}} \mathbf{w}) &\leq \nabla \ell_t(\boldsymbol{\phi}^{\mathsf{T}} \mathbf{u})^{\mathsf{T}} (\mathbf{u} - \mathbf{w}) - \frac{\sigma_t}{2} \left(\nabla \ell_t(\boldsymbol{\phi}^{\mathsf{T}} \mathbf{u})^{\mathsf{T}} (\mathbf{w} - \mathbf{u})\right)^2 \\
&= \ell_t'(\boldsymbol{\phi}^{\mathsf{T}} \mathbf{u}) \boldsymbol{\phi}_t^{\mathsf{T}} (\mathbf{u} - \mathbf{w}) - \frac{\sigma_t}{2} \left(\ell_t'(\boldsymbol{\phi}^{\mathsf{T}} \mathbf{u}) \boldsymbol{\phi}_t^{\mathsf{T}} (\mathbf{w} - \mathbf{u})\right)^2.
\end{aligned}
$$

**Definition of the epochs** We define the epochs $j \in [J]$ that are separated by $z_t = 1$, which indicate the dictionary change. Formally $z_{t_j} = 1$, and $z_{t'} = 0$ for all $t_j < t' < t_j + h_j$ where $h_j = t_{j+1} - t_j$ is the length of the epoch. We also use $\mathcal{I}_j$ for the dictionary in the $j$-th phase, $\mathbf{\Phi}_{\mathcal{I}_j}$ for the feature matrix containing samples $\phi_i \in \mathcal{I}_j$, and $\widetilde{\mathbf{P}}_j = \mathbf{\Phi}_{\mathcal{I}_j}(\mathbf{\Phi}_{\mathcal{I}_j}^\mathsf{T}\mathbf{\Phi}_{\mathcal{I}_j})^+\mathbf{\Phi}_{\mathcal{I}_j}^\mathsf{T}$ for the projection matrix on the column span of $\mathbf{\Phi}_{\mathcal{I}_j}$. Similarly, given $\mathcal{I}_j$, we define the embedding $\widetilde{\varphi}_j(\cdot)$ such that

$$\widetilde{\varphi}_j(\mathbf{x}_t) = \widetilde{\phi}_t = \mathbf{\Sigma}_j^{-1}\mathbf{U}_j^\mathsf{T}\mathbf{\Phi}_{\mathcal{I}_j}^\mathsf{T}\phi_t \in \mathbb{R}^j$$

and $\widetilde{\phi}_t^\mathsf{T}\widetilde{\phi}_{t'} = \phi_t\widetilde{\mathbf{P}}_j\phi_t'$. Given the embedding $\widetilde{\varphi}_j$, we introduce the *restricted* RKHS $\mathcal{H}_j$ and the approximate kernel function $\widetilde{\mathcal{K}}_j(\cdot,\cdot)$. Note that although the mapping $\widetilde{\varphi}_j$ changes across different epochs, it is *unique* for a fixed round $t$, therefore we simply write $\widetilde{\phi}_t$ instead of the more explicit $\widetilde{\phi}_t^j$.

**Regret decomposition** We introduce two intermediate comparators: $\overline{\mathbf{w}}_t$ and $\widehat{\mathbf{w}}_t$. First, $\widehat{\mathbf{w}}_j$ is the best fixed solution within the epoch for the dictionary of the epoch and the related $\mathcal{H}_j$. Second, $\widehat{\mathbf{w}}_j$ is the best fixed solution within the epoch over the whole space $\mathcal{H}$. Notice, neither $\widehat{\mathbf{w}}_j$ nor $\widehat{\mathbf{w}}_j$ change *during the same epoch.* Formally, for all $t'$ such that $t_j < t' \leq t_j + h_j$ we have $\overline{\mathbf{w}}_{t'} = \overline{\mathbf{w}}_j$ and $\widehat{\mathbf{w}}_{t'} = \widehat{\mathbf{w}}_j$, defined as

$$\overline{\mathbf{w}}_j = \arg\min_{\mathbf{w} \in \mathcal{H}_j} \sum_{t=t_j}^{t_{j+1}-1} \ell_t(\phi_t^\mathsf{T}\mathbf{w}) + \alpha\|\mathbf{w}\|^2, \quad \text{and} \quad \widehat{\mathbf{w}}_j = \arg\min_{\mathbf{w} \in \mathcal{H}} \sum_{t=t_j}^{t_{j+1}-1} \ell_t(\phi_t^\mathsf{T}\mathbf{w}) + \alpha\|\mathbf{w}\|^2.$$

Using the above intermediate comparators now split the regret into three parts, one per epoch,

$$\sum_{t=1}^{T} \ell_t(\phi_t^\mathsf{T}\widetilde{\mathbf{w}}_t) - \ell_t(\phi_t^\mathsf{T}\mathbf{w}) = \sum_{t=1}^{T} \ell_t(\phi_t^\mathsf{T}\widetilde{\mathbf{w}}_t) - \ell_t(\phi_t^\mathsf{T}\overline{\mathbf{w}}_t) + \ell_t(\phi_t^\mathsf{T}\overline{\mathbf{w}}_t) - \ell_t(\phi_t^\mathsf{T}\widehat{\mathbf{w}}_t) + \ell_t(\phi_t^\mathsf{T}\widehat{\mathbf{w}}_t) - \ell_t(\phi_t^\mathsf{T}\mathbf{w})$$

$$= \sum_{j=1}^{J}\sum_{t=t_j}^{t_{j+1}-1} \ell_t(\phi_t^\mathsf{T}\widetilde{\mathbf{w}}_t) - \ell_t(\phi_t^\mathsf{T}\overline{\mathbf{w}}_t) + \ell_t(\phi_t^\mathsf{T}\overline{\mathbf{w}}_t) - \ell_t(\phi_t^\mathsf{T}\widehat{\mathbf{w}}_t) + \ell_t(\phi_t^\mathsf{T}\widehat{\mathbf{w}}_t) - \ell_t(\phi_t^\mathsf{T}\mathbf{w})$$

$$= \sum_{j=1}^{J}\sum_{t=t_j}^{t_{j+1}-1} \ell_t(\phi_t^\mathsf{T}\widetilde{\mathbf{w}}_t) - \ell_t(\phi_t^\mathsf{T}\overline{\mathbf{w}}_j) + \ell_t(\phi_t^\mathsf{T}\overline{\mathbf{w}}_j) - \ell_t(\phi_t^\mathsf{T}\widehat{\mathbf{w}}_j) + \ell_t(\phi_t^\mathsf{T}\widehat{\mathbf{w}}_j) - \ell_t(\phi_t^\mathsf{T}\mathbf{w})$$

$$= \sum_{j=1}^{J} \underbrace{\left(\sum_{t=t_j}^{t_{j+1}-1} \ell_t(\phi_t^\mathsf{T}\widetilde{\mathbf{w}}_t) - \ell_t(\phi_t^\mathsf{T}\overline{\mathbf{w}}_j)\right)}_{A_j} + \underbrace{\left(\sum_{t=t_j}^{t_{j+1}-1} \ell_t(\phi_t^\mathsf{T}\overline{\mathbf{w}}_j) - \ell_t(\phi_t^\mathsf{T}\widehat{\mathbf{w}}_j)\right)}_{B_j} + \underbrace{\left(\sum_{t=t_j}^{t_{j+1}-1} \ell_t(\phi_t^\mathsf{T}\widehat{\mathbf{w}}_j) - \ell_t(\phi_t^\mathsf{T}\mathbf{w})\right)}_{C_j}.$$

Fix an epoch $j$. We now bound $A_j$, $B_j$, and $C_j$ separately.

**Bounding $A_j$** Since $\overline{\mathbf{w}}_j \in \mathcal{H}_j$, projecting $\overline{\mathbf{w}}_j$ to $\mathcal{H}_j$ won't change it, i.e., $\overline{\mathbf{w}}_j = \widetilde{\mathbf{P}}_j\overline{\mathbf{w}}_j$ and

$$\phi_t^\mathsf{T}\overline{\mathbf{w}}_j = \phi_t^\mathsf{T}\widetilde{\mathbf{P}}_j\overline{\mathbf{w}}_j = \phi_t^\mathsf{T}\mathbf{\Phi}_{\mathcal{I}_j}\mathbf{U}_j\mathbf{\Sigma}_j^{-1}\mathbf{\Sigma}_j^{-1}\mathbf{U}_j^\mathsf{T}\mathbf{\Phi}_{\mathcal{I}_j}^\mathsf{T}\overline{\mathbf{w}}_j = \widetilde{\phi}_t^\mathsf{T}\overline{\boldsymbol{\omega}}_j$$

with $\overline{\boldsymbol{\omega}}_j = \mathbf{\Sigma}_j^{-1}\mathbf{U}_j^\mathsf{T}\mathbf{\Phi}_{\mathcal{I}_j}^\mathsf{T}\overline{\mathbf{w}}_j$. Remembering that $\widetilde{\mathbf{w}}_t = \mathbf{\Phi}_{\mathcal{I}_j}\mathbf{U}_j\mathbf{\Sigma}_j^{-1}\widetilde{\boldsymbol{\omega}}_t$ for all $t$ in the epoch

$$\ell_t\left(\phi_t^\mathsf{T}\widetilde{\mathbf{w}}_t\right) - \ell_t\left(\phi_t^\mathsf{T}\overline{\mathbf{w}}_j\right) = \ell_t\left(\phi_t^\mathsf{T}\mathbf{\Phi}_{\mathcal{I}_j}\mathbf{U}_j\mathbf{\Sigma}_j^{-1}\widetilde{\boldsymbol{\omega}}_t\right) - \ell_t\left(\phi_t^\mathsf{T}\widetilde{\mathbf{P}}_j\overline{\mathbf{w}}_j\right) = \ell_t\left(\widetilde{\phi}_t^\mathsf{T}\widetilde{\boldsymbol{\omega}}_t\right) - \ell_t\left(\widetilde{\phi}_t^\mathsf{T}\overline{\boldsymbol{\omega}}_j\right).$$

Now, using Assumption 2, we get

$$\ell_t(\widetilde{\phi}_t^\mathsf{T}\widetilde{\boldsymbol{\omega}}_t) - \ell_t(\widetilde{\phi}_t^\mathsf{T}\overline{\boldsymbol{\omega}}_j) \leq \widetilde{\mathbf{g}}_t^\mathsf{T}(\widetilde{\boldsymbol{\omega}}_t - \overline{\boldsymbol{\omega}}_j) - \frac{\sigma_t}{2}(\widetilde{\mathbf{g}}_t^\mathsf{T}(\widetilde{\boldsymbol{\omega}}_t - \overline{\boldsymbol{\omega}}_j))^2 \tag{7}$$

and due to the update rules and the contracting property of projections [8],

$$\|\widetilde{\boldsymbol{\omega}}_t - \overline{\boldsymbol{\omega}}_j\|_{\widetilde{\mathbf{A}}_{t-1}}^2 \leq \|\widetilde{\boldsymbol{\omega}}_{t-1} - \widetilde{\mathbf{A}}_{t-1}^{-1}\widetilde{\mathbf{g}}_{t-1} - \overline{\boldsymbol{\omega}}_j\|_{\widetilde{\mathbf{A}}_{t-1}}^2$$

$$= \|\widetilde{\boldsymbol{\omega}}_{t-1} - \overline{\boldsymbol{\omega}}_j\|_{\widetilde{\mathbf{A}}_{t-1}}^2 - 2\widetilde{\mathbf{g}}_{t-1}^\mathsf{T}\widetilde{\mathbf{A}}_{t-1}^{-1}\widetilde{\mathbf{A}}_{t-1}(\widetilde{\boldsymbol{\omega}}_{t-1} - \overline{\boldsymbol{\omega}}_j) + \|\widetilde{\mathbf{A}}_{t-1}^{-1}\widetilde{\mathbf{g}}_{t-1}\|_{\widetilde{\mathbf{A}}_{t-1}}^2$$

$$= \|\widetilde{\boldsymbol{\omega}}_{t-1} - \overline{\boldsymbol{\omega}}_j\|_{\widetilde{\mathbf{A}}_{t-1}}^2 - 2\widetilde{\mathbf{g}}_{t-1}^\mathsf{T}(\widetilde{\boldsymbol{\omega}}_{t-1} - \overline{\boldsymbol{\omega}}_j) + \widetilde{\mathbf{g}}_{t-1}^\mathsf{T}\widetilde{\mathbf{A}}_{t-1}^{-1}\mathbf{g}_{t-1}$$

except for last round of epoch $j$, i.e., round $t_{j+1} - 1$, where due to the reset we cannot use the update rule, the we need to treat separately. Therefore, for any round $t = t_j, t_j + 1, \ldots, t_{j+1} - 2$ of epoch $j$, we have that

$$2\widetilde{\mathbf{g}}_t^\mathsf{T}(\widetilde{\boldsymbol{\omega}}_t - \overline{\boldsymbol{\omega}}_j) \le \|\widetilde{\boldsymbol{\omega}}_t - \overline{\boldsymbol{\omega}}_j\|_{\widetilde{\mathbf{A}}_t}^2 - \|\widetilde{\boldsymbol{\omega}}_{t+1} - \overline{\boldsymbol{\omega}}_j\|_{\widetilde{\mathbf{A}}_t}^2 + \widetilde{\mathbf{g}}_t^\mathsf{T}\widetilde{\mathbf{A}}_t^{-1}\mathbf{g}_t. \tag{8}$$

Using the upper bound on the loss difference (7), we get

$$\sum_{t=t_j}^{t_{j+1}-1} \ell_t \left(\boldsymbol{\phi}_t^\mathsf{T}\widetilde{\mathbf{w}}_t\right) - \ell_t \left(\boldsymbol{\phi}_t^\mathsf{T}\overline{\mathbf{w}}_t\right)$$

$$= \ell_{t_{j+1}-1}\left(\boldsymbol{\phi}_{t_{j+1}-1}^\mathsf{T}\widetilde{\mathbf{w}}_{t_{j+1}-1}\right) - \ell_{t_{j+1}-1}\left(\boldsymbol{\phi}_{t_{j+1}-1}^\mathsf{T}\overline{\mathbf{w}}_{t_{j+1}-1}\right) + \sum_{t=t_j}^{t_{j+1}-2} \ell_t \left(\boldsymbol{\phi}_t^\mathsf{T}\widetilde{\mathbf{w}}_t\right) - \ell_t \left(\boldsymbol{\phi}_t^\mathsf{T}\overline{\mathbf{w}}_t\right)$$

$$= R_j + \sum_{t=t_j}^{t_{j+1}-2} \ell_t \left(\widetilde{\boldsymbol{\phi}}_t^\mathsf{T}\widetilde{\boldsymbol{\omega}}_t\right) - \ell_t \left(\widetilde{\boldsymbol{\phi}}_t^\mathsf{T}\overline{\boldsymbol{\omega}}_j\right)$$

$$\le R_j + \sum_{t=t_j}^{t_{j+1}-2} \widetilde{\mathbf{g}}_t^\mathsf{T} (\widetilde{\boldsymbol{\omega}}_t - \overline{\boldsymbol{\omega}}_j) - \frac{\sigma_t}{2} \left(\widetilde{\mathbf{g}}_t^\mathsf{T} (\widetilde{\boldsymbol{\omega}}_t - \overline{\boldsymbol{\omega}}_j)\right)^2,$$

with $R_j \overset{\text{def}}{=} \ell_{t_{j+1}-1}\left(\boldsymbol{\phi}_{t_{j+1}-1}^\mathsf{T}\widetilde{\mathbf{w}}_{t_{j+1}-1}\right) - \ell_{t_{j+1}-1}\left(\boldsymbol{\phi}_{t_{j+1}-1}^\mathsf{T}\overline{\mathbf{w}}_{t_{j+1}-1}\right)$ that corresponds to the regret of the last round of the epoch that we need to treat separately. Now for the all other rounds,

$$\sum_{t=t_j}^{t_{j+1}-2} \widetilde{\mathbf{g}}_t^\mathsf{T} (\widetilde{\boldsymbol{\omega}}_t - \overline{\boldsymbol{\omega}}_j) - \frac{\sigma_t}{2} \left(\widetilde{\mathbf{g}}_t^\mathsf{T} (\widetilde{\boldsymbol{\omega}}_t - \overline{\boldsymbol{\omega}}_j)\right)^2$$

$$\le \sum_{t=t_j}^{t_{j+1}-2} \|\widetilde{\boldsymbol{\omega}}_t - \overline{\boldsymbol{\omega}}_j\|_{\widetilde{\mathbf{A}}_t}^2 - \|\widetilde{\boldsymbol{\omega}}_{t+1} - \overline{\boldsymbol{\omega}}_j\|_{\widetilde{\mathbf{A}}_t}^2 + \widetilde{\mathbf{g}}_t^\mathsf{T}\widetilde{\mathbf{A}}_t^{-1}\mathbf{g}_t - \|\widetilde{\boldsymbol{\omega}}_t - \overline{\boldsymbol{\omega}}_j\|_{\frac{\sigma_t}{2}\widetilde{\mathbf{g}}_t\widetilde{\mathbf{g}}_t^\mathsf{T}}^2$$

$$= -\|\widetilde{\boldsymbol{\omega}}_{t_{j+1}-1} - \overline{\boldsymbol{\omega}}_j\|_{\widetilde{\mathbf{A}}_{t_{j+1}-1}}^2 + \|\widetilde{\boldsymbol{\omega}}_{t_j} - \overline{\boldsymbol{\omega}}_j\|_{\widetilde{\mathbf{A}}_{t_j}}^2 + \widetilde{\mathbf{g}}_{t_j}^\mathsf{T}\widetilde{\mathbf{A}}_{t_j}^{-1}\mathbf{g}_{t_j} - \|\widetilde{\boldsymbol{\omega}}_{t_j} - \overline{\boldsymbol{\omega}}_j\|_{\frac{\sigma_{t_j}}{2}\widetilde{\mathbf{g}}_{t_j}\widetilde{\mathbf{g}}_{t_j}^\mathsf{T}}^2$$

$$+ \sum_{t=t_j+1}^{t_{j+1}-2} \|\widetilde{\boldsymbol{\omega}}_t - \overline{\boldsymbol{\omega}}_j\|_{\widetilde{\mathbf{A}}_t}^2 - \|\widetilde{\boldsymbol{\omega}}_t - \overline{\boldsymbol{\omega}}_j\|_{\widetilde{\mathbf{A}}_{t-1}}^2 + \widetilde{\mathbf{g}}_t^\mathsf{T}\widetilde{\mathbf{A}}_t^{-1}\mathbf{g}_t - \|\widetilde{\boldsymbol{\omega}}_t - \overline{\boldsymbol{\omega}}_j\|_{\frac{\sigma_t}{2}\widetilde{\mathbf{g}}_t\widetilde{\mathbf{g}}_t^\mathsf{T}}^2$$

Now we treat the terms above equation separately. First, since $\widetilde{\mathbf{A}}_t = \widetilde{\mathbf{A}}_{t-1} + \frac{\sigma_t}{2}\widetilde{\mathbf{g}}_t\widetilde{\mathbf{g}}_t^\mathsf{T}$, we have $\|\widetilde{\boldsymbol{\omega}}_t - \overline{\boldsymbol{\omega}}_j\|_{\widetilde{\mathbf{A}}_{t-1}}^2 + \|\widetilde{\boldsymbol{\omega}}_t - \overline{\boldsymbol{\omega}}_j\|_{\frac{\sigma_t}{2}\widetilde{\mathbf{g}}_t\widetilde{\mathbf{g}}_t^\mathsf{T}}^2 = \|\widetilde{\boldsymbol{\omega}}_t - \overline{\boldsymbol{\omega}}_j\|_{\widetilde{\mathbf{A}}_t}^2$. Second, by Algorithm 1, we know that at the beginning of each epoch, $\widetilde{\mathbf{A}}_{t_j} = \alpha\mathbf{I} + \widetilde{\mathbf{g}}_{t_j}\widetilde{\mathbf{g}}_{t_j}^\mathsf{T}$. This also helps us to bound the term $\widetilde{\mathbf{g}}_{t_j}^\mathsf{T}\widetilde{\mathbf{A}}_{t_j}^{-1}\mathbf{g}_{t_j}$ as

$$\widetilde{\mathbf{g}}_{t_j}^\mathsf{T}\widetilde{\mathbf{A}}_{t_j}^{-1}\mathbf{g}_{t_j} = \widetilde{\mathbf{g}}_{t_j}^\mathsf{T}\left(\alpha\mathbf{I} + \frac{\sigma_{t_j}}{2}\widetilde{\mathbf{g}}_{t_j}\widetilde{\mathbf{g}}_{t_j}^\mathsf{T}\right)^{-1}\mathbf{g}_{t_j} = \frac{\widetilde{\mathbf{g}}_{t_j}^\mathsf{T}\widetilde{\mathbf{g}}_{t_j}}{\alpha + \frac{\sigma_{t_j}}{2}\widetilde{\mathbf{g}}_{t_j}^\mathsf{T}\widetilde{\mathbf{g}}_{t_j}} \le \frac{\widetilde{\mathbf{g}}_{t_j}^\mathsf{T}\widetilde{\mathbf{g}}_{t_j}}{\alpha} = \frac{\left(\ell_{t_j}'\left(\widetilde{y}_{t_j}\right)\right)^2 \widetilde{\boldsymbol{\phi}}_{t_j}^\mathsf{T}\widetilde{\boldsymbol{\phi}}_{t_j}}{\alpha}$$

$$\le \frac{L^2\widetilde{\boldsymbol{\phi}}_{t_j}^\mathsf{T}\widetilde{\boldsymbol{\phi}}_{t_j}}{\alpha} = \frac{L^2\boldsymbol{\phi}_{t_j}^\mathsf{T}\widetilde{\mathbf{P}}_j\boldsymbol{\phi}_{t_j}}{\alpha} \le \frac{L^2\boldsymbol{\phi}_{t_j}^\mathsf{T}\boldsymbol{\phi}_{t_j}}{\alpha} \le \frac{L^2}{\alpha}.$$

By Algorithm 1, we also know that at the beginning of each epoch $\widetilde{\boldsymbol{\omega}}_{t_j} = \mathbf{0}$ which helps us to bound the two terms outside of the summation as

$$\|\widetilde{\boldsymbol{\omega}}_{t_j} - \overline{\boldsymbol{\omega}}_j\|_{\widetilde{\mathbf{A}}_{t_j}}^2 - \|\widetilde{\boldsymbol{\omega}}_{t_j} - \overline{\boldsymbol{\omega}}_j\|_{\frac{\sigma_{t_j}}{2}\widetilde{\mathbf{g}}_{t_j}\widetilde{\mathbf{g}}_{t_j}^\mathsf{T}}^2$$

$$= \|\widetilde{\boldsymbol{\omega}}_{t_j} - \overline{\boldsymbol{\omega}}_j\|_{\alpha\mathbf{I}}^2 + \|\widetilde{\boldsymbol{\omega}}_{t_j} - \overline{\boldsymbol{\omega}}_j\|_{\frac{\sigma_{t_j}}{2}\widetilde{\mathbf{g}}_{t_j}\widetilde{\mathbf{g}}_{t_j}^\mathsf{T}}^2 - \|\widetilde{\boldsymbol{\omega}}_{t_j} - \overline{\boldsymbol{\omega}}_j\|_{\frac{\sigma_{t_j}}{2}\widetilde{\mathbf{g}}_{t_j}\widetilde{\mathbf{g}}_{t_j}^\mathsf{T}}^2$$

$$= \alpha\|\widetilde{\boldsymbol{\omega}}_{t_j} - \overline{\boldsymbol{\omega}}_j\|_2^2 = \alpha\|\overline{\boldsymbol{\omega}}_j\|_2^2.$$

Altogether, we combine the upper bounds of the terms to

$$\sum_{t=t_j}^{t_{j+1}-1} \ell_t(\boldsymbol{\phi}_t^\mathsf{T}\widetilde{\mathbf{w}}_t) - \ell_t(\boldsymbol{\phi}_t^\mathsf{T}\overline{\mathbf{w}}_t) \le \alpha\|\overline{\boldsymbol{\omega}}_j\|^2 + R_j + \frac{L^2}{\alpha} - \|\widetilde{\boldsymbol{\omega}}_{t_{j+1}-1} - \overline{\boldsymbol{\omega}}_j\|_{\widetilde{\mathbf{A}}_{t_{j+1}-1}}^2 + \sum_{t=t_j+1}^{t_{j+1}-2} \widetilde{\mathbf{g}}_t^\mathsf{T}\widetilde{\mathbf{A}}_t^{-1}\mathbf{g}_t.$$

Using the result of [8] we can upper bound the sum of the quadratic forms as

$$\sum_{t=t_j}^{t_{j+1}-2} \widetilde{\mathbf{g}}_t^\intercal \widetilde{\mathbf{A}}_t^{-1} \widetilde{\mathbf{g}}_t = \sum_{t=t_j}^{t_{j+1}-2} \widetilde{\mathbf{g}}_t^\intercal \left( \sum_{s=t_j}^{t} \tfrac{\sigma_s}{2} \widetilde{\mathbf{g}}_s \widetilde{\mathbf{g}}_s^\intercal + \alpha \mathbf{I} \right)^{-1} \widetilde{\mathbf{g}}_t$$

$$= \sum_{t=t_j}^{t_{j+1}-2} \frac{2}{\sigma_t} \left( \sqrt{\sigma_t/2\alpha} \cdot \widetilde{\mathbf{g}}_t^\intercal \right) \left( \sum_{s=t_j}^{t} \tfrac{\sigma_s}{2} \widetilde{\mathbf{g}}_s \widetilde{\mathbf{g}}_s^\intercal/\alpha + \mathbf{I} \right)^{-1} \left( \sqrt{\sigma_t/2\alpha} \cdot \widetilde{\mathbf{g}}_t \right)$$

$$\leq \frac{2}{\sigma_{\min}} \log \left( \mathrm{Det} \left( \widetilde{\mathbf{G}}_j \widetilde{\mathbf{G}}_j^\intercal/\alpha + \mathbf{I} \right) \right),$$

where $\widetilde{\mathbf{G}}_j$ is the $j \times h_j$ matrix with $\sqrt{\sigma_t/2} \cdot \widetilde{\mathbf{g}}_t$ columns. Let $\widetilde{\mathbf{D}}_j$ be the $h_j \times h_j$ diagonal matrix with $\dot{g}_t \sqrt{\sigma_t/2}$ on the diagonal and $\mathbf{\Phi}_j$ (resp., $\widetilde{\mathbf{\Phi}}_j$) the matrix with $\phi_t$ (resp., $\tilde{\phi}_t$) as columns for $t_j \leq t < t_{j+1}$ We can rewrite $\widetilde{\mathbf{G}}_j = \widetilde{\mathbf{\Phi}}_j \widetilde{\mathbf{D}}_j = \mathbf{\Sigma}_j^{-1} \mathbf{U}_j^\intercal \mathbf{\Phi}_{\mathcal{I}_j}^\intercal \mathbf{\Phi}_j \widetilde{\mathbf{D}}_j$. We also have

$$\widetilde{\mathbf{G}}_j^\intercal \widetilde{\mathbf{G}}_j = \widetilde{\mathbf{D}}_j \mathbf{\Phi}_j^\intercal \mathbf{\Phi}_{\mathcal{I}_j} \mathbf{U}_j^\intercal \mathbf{\Sigma}_j^{-1} \mathbf{\Sigma}_j^{-1} \mathbf{U}_j^\intercal \mathbf{\Phi}_{\mathcal{I}_j}^\intercal \mathbf{\Phi}_j \widetilde{\mathbf{D}}_j = \widetilde{\mathbf{D}}_j \mathbf{\Phi}_j^\intercal \widetilde{\mathbf{P}}_j \mathbf{\Phi}_j \widetilde{\mathbf{D}}_j \preceq \widetilde{\mathbf{D}}_j \mathbf{\Phi}_{\mathcal{I}_j}^\intercal \mathbf{\Phi}_{\mathcal{I}_j} \widetilde{\mathbf{D}}_j,$$

since $\|\widetilde{\mathbf{P}}_j\| \leq 1$ because $\widetilde{\mathbf{P}}_j$ is a projection matrix. Knowing that $\mathrm{Det}(\mathbf{A}) \leq \mathrm{Det}(\mathbf{B})$ whenever $\mathbf{A} \preceq \mathbf{B}$, together with Sylvester's determinant identity, we get that

$$\mathrm{Det}(\widetilde{\mathbf{G}}_j \widetilde{\mathbf{G}}_j^\intercal/\alpha + \mathbf{I}) \leq \mathrm{Det}(\widetilde{\mathbf{D}}_j \mathbf{\Phi}_j^\intercal \mathbf{\Phi}_j \widetilde{\mathbf{D}}_j/\alpha + \mathbf{I}) = \prod_{t=1}^{h_j} (\lambda_t/\alpha + 1),$$

where $\lambda_t$ are the eigenvalues of $\widetilde{\mathbf{D}}_j \mathbf{\Phi}_{\mathcal{I}_j}^\intercal \mathbf{\Phi}_{\mathcal{I}_j} \widetilde{\mathbf{D}}_j = \widetilde{\mathbf{D}}_j \mathbf{K}_j \widetilde{\mathbf{D}}_j = \overline{\mathbf{K}}_j$ and $\mathbf{K}_j$ is the kernel matrix between the samples in epoch $j$. Using the result of Calandriello et al. [2, Lemma 3] we can further bound the expression above as

$$\log \left( \prod_{t=1}^{h_j} \lambda_t/\alpha + 1 \right) \leq 2 d_{\mathrm{eff}}^j \left( \frac{\alpha}{\sigma_{\min} L^2} \right) \log \left( 2\sigma_{\min} L^2 \, \mathrm{Tr}(\mathbf{K}_j)/\alpha \right)$$

$$\leq 2 d_{\mathrm{eff}}^T \left( \frac{\alpha}{\sigma_{\min} L^2} \right) \log \left( 2\sigma_{\min} L^2 \, \mathrm{Tr}(\mathbf{K}_t)/\alpha \right).$$

Putting it all together, and using $\|\overline{\boldsymbol{\omega}}_j\|_2^2 = \|\mathbf{\Sigma}_j^{-1} \mathbf{U}_j \mathbf{\Phi}_{\mathcal{I}_j}^\intercal \overline{\mathbf{w}}_j\|_2^2 = \overline{\mathbf{w}}_j^\intercal \widetilde{\mathbf{P}}_j \overline{\mathbf{w}}_j = \|\overline{\mathbf{w}}_j\|_2^2$ we get

$$A_j \leq \frac{4}{\sigma_{\min}} d_{\mathrm{eff}}^T \left( \frac{\alpha}{\sigma_{\min} L^2} \right) \log \left( 2\sigma_{\min} L^2 \, \mathrm{Tr}(\mathbf{K}_j)/\alpha \right) + \alpha \|\overline{\mathbf{w}}_j\|_2^2$$

$$+ R_j + \frac{L^2}{\alpha} - \|\widetilde{\boldsymbol{\omega}}_{t_{j+1}-1} - \overline{\boldsymbol{\omega}}_j\|_{\widetilde{\mathbf{A}}_{t_{j+1}-1}}^2.$$

**Bounding $B_j$**    We begin by adding and subtracting $\alpha \|\overline{\mathbf{w}}_j\|^2$ and $\alpha \|\widehat{\mathbf{w}}_j\|^2$

$$\sum_{t=t_j}^{t_{j+1}-1} \ell_t(\phi_t^\intercal \overline{\mathbf{w}}_j) - \ell_t(\phi_t^\intercal \widehat{\mathbf{w}}_j) = \left( \sum_{t=t_j}^{t_{j+1}-1} \ell_t(\phi_t^\intercal \overline{\mathbf{w}}_j) \right) - \left( \sum_{t=t_j}^{t_{j+1}-1} \ell_t(\phi_t^\intercal \widehat{\mathbf{w}}_j) \right)$$

$$= \alpha \|\widehat{\mathbf{w}}_j\|^2 - \alpha \|\overline{\mathbf{w}}_j\|^2 + \left( \sum_{t=t_j}^{t_{j+1}-1} \ell_t(\phi_t^\intercal \overline{\mathbf{w}}_j) + \alpha \|\overline{\mathbf{w}}_j\|^2 \right) - \left( \sum_{t=t_j}^{t_{j+1}-1} \ell_t(\phi_t^\intercal \widehat{\mathbf{w}}_j) + \alpha \|\widehat{\mathbf{w}}_j\|^2 \right)$$

We will now apply the following result from Xu et al. [20].

**Proposition 2** (Xu et al. [20, Lemma 2]). *Suppose the loss functions $\ell_t$ are $L$-Lipschitz continuous, and $\overline{\mathbf{w}}_j = \widetilde{\mathbf{P}}_j \overline{\mathbf{w}}_j = \mathbf{\Phi}_{\mathcal{I}_j} \mathbf{U}_j \mathbf{\Sigma}_j^{-1/2} \overline{\boldsymbol{\omega}}_j$. We have*

$$\frac{1}{h_j} \sum_{t=t_j}^{t_{j+1}-1} \ell_t(\phi_t^\intercal \overline{\mathbf{w}}_j) + \frac{\alpha}{2} \|\overline{\mathbf{w}}_j\|^2 \leq \frac{1}{h_j} \sum_{t=t_j}^{t_{j+1}-1} \ell_t(\phi_t^\intercal \widehat{\mathbf{w}}_j) + \frac{\alpha}{2} \|\widehat{\mathbf{w}}_j\|^2 + \frac{L^2}{2\alpha h_j} \|\mathbf{\Phi}_j - \widetilde{\mathbf{P}}_j \mathbf{\Phi}_j\|_2^2.$$

First, it is important to quantify the last term in Proposition 2,

$$\|\mathbf{\Phi}_j - \widetilde{\mathbf{P}}_j\mathbf{\Phi}_j\|_2^2 = \lambda_{\max}\left(\left(\mathbf{\Phi}_j - \widetilde{\mathbf{P}}_j\mathbf{\Phi}_j\right)^\top\left(\mathbf{\Phi}_j - \widetilde{\mathbf{P}}_j\mathbf{\Phi}_j\right)\right)$$

$$= \lambda_{\max}\left(\mathbf{\Phi}_j^\top\mathbf{\Phi}_j - 2\mathbf{\Phi}_j^\top\widetilde{\mathbf{P}}_j\mathbf{\Phi}_j + \mathbf{\Phi}_j^\top\widetilde{\mathbf{P}}_j\widetilde{\mathbf{P}}_j\mathbf{\Phi}_j\right)$$

$$= \lambda_{\max}\left(\mathbf{\Phi}_j^\top\mathbf{\Phi}_j - \mathbf{\Phi}_j^\top\widetilde{\mathbf{P}}_j\mathbf{\Phi}_j\right) = \lambda_{\max}\left(\mathbf{K}_j - \widetilde{\mathbf{K}}_j\right) \le \frac{\gamma\varepsilon}{1-\varepsilon},$$

when in the last step we applied Proposition 1 that bounds the quality of the approximation. In order to apply Proposition 2 we also need to rescale $B_j$,

$$\sum_{t=t_j}^{t_{j+1}-1} \ell_t(\boldsymbol{\phi}_t^\top\overline{\mathbf{w}}_j) + \alpha\|\overline{\mathbf{w}}_j\|^2 = h_j\left(\frac{1}{h_j}\sum_{t=t_j}^{t_{j+1}-1}\ell_t(\boldsymbol{\phi}_t^\top\overline{\mathbf{w}}_j) + \frac{\alpha}{2}\frac{2}{h_j}\|\overline{\mathbf{w}}_j\|^2\right)$$

$$\le h_j\left(\frac{1}{h_j}\sum_{t=t_j}^{t_{j+1}-1}\ell_t(\boldsymbol{\phi}_t^\top\widehat{\mathbf{w}}_j) + \frac{\alpha}{2}\frac{2}{h_j}\|\widehat{\mathbf{w}}_j\|^2 + \frac{L^2 h_j}{4\alpha h_j}\|\mathbf{\Phi}_j - \widetilde{\mathbf{P}}_j\mathbf{\Phi}_j\|_2^2\right)$$

$$= \sum_{t=t_j}^{t_{j+1}-1}\ell_t(\boldsymbol{\phi}_t^\top\widehat{\mathbf{w}}_j) + \alpha\|\widehat{\mathbf{w}}_j\|^2 + \frac{L^2 h_j}{4\alpha}\|\mathbf{K}_j - \widetilde{\mathbf{K}}_j\|_2^2$$

$$\le \sum_{t=t_j}^{t_{j+1}-1}\ell_t(\boldsymbol{\phi}_t^\top\widehat{\mathbf{w}}_j) + \alpha\|\widehat{\mathbf{w}}_j\|^2 + \frac{L^2\varepsilon}{4(1-\varepsilon)}\frac{h_j\gamma}{\alpha}.$$

Therefore, the difference of the regularized losses for the best solution within epoch $j$ when considering the whole space $\mathcal{H}$ versus subspace $\mathcal{H}_j$ is bounded as

$$\left(\sum_{t=t_j}^{t_{j+1}-1}\ell_t(\boldsymbol{\phi}_t^\top\overline{\mathbf{w}}_j) + \alpha\|\overline{\mathbf{w}}_j\|\right) - \left(\sum_{t=t_j}^{t_{j+1}-1}\ell_t(\boldsymbol{\phi}_t^\top\widehat{\mathbf{w}}_j) + \alpha\|\widehat{\mathbf{w}}_j\|\right) \le \frac{L^2\varepsilon}{4(1-\varepsilon)}\frac{h_j\gamma}{\alpha}$$

and therefore their unregularized counterparts are bounded as

$$B_j \le \alpha\|\widehat{\mathbf{w}}_j\|^2 - \alpha\|\overline{\mathbf{w}}_j\|^2 + \frac{L^2\varepsilon}{4(1-\varepsilon)}\frac{h_j\gamma}{\alpha}.$$

**Bounding $C_j$**    Similarly as for $B_j$, we add and subtract the regularizers,

$$\sum_{t=t_j}^{t_{j+1}-1}\ell_t(\boldsymbol{\phi}_t^\top\widehat{\mathbf{w}}_j) - \ell_t(\boldsymbol{\phi}_t^\top\mathbf{w}) = \left(\sum_{t=t_j}^{t_{j+1}-1}\ell_t(\boldsymbol{\phi}_t^\top\widehat{\mathbf{w}}_j)\right) - \left(\sum_{t=t_j}^{t_{j+1}-1}\ell_t(\boldsymbol{\phi}_t^\top\mathbf{w})\right)$$

$$= \alpha\|\mathbf{w}\|^2 - \alpha\|\widehat{\mathbf{w}}_j\|^2 + \left(\sum_{t=t_j}^{t_{j+1}-1}\ell_t(\boldsymbol{\phi}_t^\top\widehat{\mathbf{w}}_j) + \alpha\|\widehat{\mathbf{w}}_j\|^2\right) - \left(\sum_{t=t_j}^{t_{j+1}-1}\ell_t(\boldsymbol{\phi}_t^\top\mathbf{w}) + \alpha\|\mathbf{w}\|^2\right).$$

By the definition of $\widehat{\mathbf{w}}_j$ as a minimizer, we have that the difference between the summations is negative or zero. Therefore, term $C_j$ is trivially bounded as

$$C_j \le \alpha\|\mathbf{w}\|^2 - \alpha\|\widehat{\mathbf{w}}_j\|^2.$$

**Bounding the regret:** We put all the bounds on decomposed regret together:

$$\sum_{t=1}^{T}\ell_t(\boldsymbol{\phi}_t^\mathsf{T}\widetilde{\mathbf{w}}_t) - \ell_t(\boldsymbol{\phi}_t^\mathsf{T}\mathbf{w}) = \sum_{j=1}^{J} A_j + B_j + C_j$$

$$\leq \sum_{j=1}^{J} \frac{4}{\sigma_{\min}} d_{\text{eff}}^T\left(\frac{\alpha}{\sigma_{\min}L^2}\right) \log\left(2\sigma_{\min}L^2\,\mathrm{Tr}(\mathbf{K}_j)/\alpha\right) + \alpha\|\overline{\mathbf{w}}_j\|_2^2 + R_j + \frac{L^2}{\alpha} - \|\widetilde{\boldsymbol{\omega}}_{t_{j+1}-1} - \overline{\boldsymbol{\omega}}_j\|_{\widetilde{\mathbf{A}}_{t_{j+1}-1}}^2$$

$$+ \alpha\|\widehat{\mathbf{w}}_j\|^2 - \alpha\|\overline{\mathbf{w}}_j\|^2 + \frac{L^2\varepsilon}{4(1-\varepsilon)}\frac{h_j\gamma}{\alpha} + \alpha\|\mathbf{w}\|^2 - \alpha\|\widehat{\mathbf{w}}_j\|^2$$

$$= \left(\sum_{j=1}^{J} \frac{4}{\sigma_{\min}} d_{\text{eff}}^T\left(\frac{\alpha}{\sigma_{\min}L^2}\right) \log\left(2\sigma_{\min}L^2\,\mathrm{Tr}\left(\mathbf{K}_j\right)/\alpha\right)\right) + \left(\sum_{j=1}^{J} \frac{L^2\varepsilon}{4(1-\varepsilon)}\frac{h_j\gamma}{\alpha}\right) + \frac{JL^2}{\alpha} + J\alpha\|\mathbf{w}\|$$

$$+ \sum_{j=1}^{J} R_j - \|\widetilde{\boldsymbol{\omega}}_{t_{j+1}-1} - \overline{\boldsymbol{\omega}}_j\|_{\widetilde{\mathbf{A}}_{t_{j+1}-1}}^2$$

$$\leq \left(\sum_{j=1}^{J} \frac{4}{\sigma_{\min}} d_{\text{eff}}^T\left(\frac{\alpha}{\sigma_{\min}L^2}\right) \log\left(2\sigma_{\min}L^2 T/\alpha\right)\right) + \frac{L^2}{\alpha}\left(\frac{T\gamma\varepsilon}{4(1-\varepsilon)} + 1\right) + J\alpha\|\mathbf{w}\|$$

$$+ \sum_{j=1}^{J} R_j - \|\widetilde{\boldsymbol{\omega}}_{t_{j+1}-1} - \overline{\boldsymbol{\omega}}_j\|_{\widetilde{\mathbf{A}}_{t_{j+1}-1}}^2$$

$$\leq J\alpha\|\mathbf{w}\| + \frac{4J}{\sigma_{\min}} d_{\text{eff}}^T\left(\frac{\alpha}{\sigma_{\min}L^2}\right) \log\left(2\sigma_{\min}L^2 T/\alpha\right) + \frac{L^2}{\alpha}\left(\frac{T\gamma\varepsilon}{4(1-\varepsilon)} + 1\right) + \sum_{j=1}^{J} R_j$$

$$= 3\beta d_{\text{eff}}^T(\gamma) \log\left(2T\right) \left(\frac{4}{\sigma_{\min}} d_{\text{eff}}^T\left(\frac{\alpha}{\sigma_{\min}L^2}\right) \log\left(2\sigma_{\min}L^2 T/\alpha\right) + \alpha\|\mathbf{w}\|\right) + \frac{L^2}{\alpha}\left(\frac{T\gamma\varepsilon}{4(1-\varepsilon)} + 1\right) + \sum_{j=1}^{J} R_j$$

## C Proof of Theorem 2: Regret bound for squared loss

In the special case of *squared loss*, we can obtain a different kind of guarantee. We proceed in a similar way as in Appendix B and highlight the differences. Starting from the $A_j + B_j + C_j$ decomposition given in Appendix B, we will bound $B_j$ differently using the following result.

**Proposition 3** (Zhdanov and Kalnishkan [24, Thm. 1]). *Take a kernel $\mathcal{K}$ on a domain $\mathcal{X}$ and a parameter $\alpha > 0$. Let $\mathcal{H}$ be the RKHS for the kernel $\mathcal{K}$. For any sequence $\{(\mathbf{x}_t, y_t)\}_{t=1}^{T}$ let $\mathbf{y}_T \in \mathbb{R}^T$ be the concatenation of the $y_t$ target variables. Then*

$$\mathcal{L}_T^*(\mathcal{H}) = \min_{f\in\mathcal{H}} \left(\sum_{t=1}^{T} \left(f(\mathbf{x}_t) - y_t\right)^2 + \alpha\|f\|_\mathcal{H}^2\right) = \alpha\mathbf{y}_T^\mathsf{T}(\mathbf{K}_T + \alpha\mathbf{I})^{-1}\mathbf{y}_T.$$

**Bounding $B_j$** In our particular case, we apply Proposition 3 to the whole space $\mathcal{H}$ and all subspaces $\mathcal{H}_j$, one for each epoch $j$.

$$\mathcal{L}_j^*(\mathcal{H}) = \sum_{t=t_j}^{t_{j+1}-1} \left(\boldsymbol{\phi}_t^\mathsf{T}\widehat{\mathbf{w}}_j - y_t\right)^2 + \alpha\|\widehat{\mathbf{w}}_j\|_2^2 = \min_{\mathbf{w}\in\mathcal{H}}\left(\sum_{t=t_j}^{t_{j+1}-1} \left(\boldsymbol{\phi}_t^\mathsf{T}\mathbf{w} - y_t\right)^2 + \alpha\|\mathbf{w}\|_2^2\right) = \alpha\mathbf{y}_j^\mathsf{T}(\boldsymbol{\Phi}_j^\mathsf{T}\boldsymbol{\Phi}_j + \alpha\mathbf{I})^{-1}\mathbf{y}_j,$$

$$\mathcal{L}_j^*(\mathcal{H}_j) = \sum_{t=t_j}^{t_{j+1}-1} \left(\boldsymbol{\phi}_t^\mathsf{T}\overline{\mathbf{w}}_j - y_t\right)^2 + \alpha\|\overline{\mathbf{w}}_j\|_2^2 = \min_{\mathbf{w}\in\mathcal{H}_j}\left(\sum_{t=t_j}^{t_{j+1}-1} \left(\boldsymbol{\phi}_t^\mathsf{T}\mathbf{w} - y_t\right)^2 + \alpha\|\mathbf{w}\|_2^2\right) = \alpha\mathbf{y}_j^\mathsf{T}(\boldsymbol{\Phi}_j^\mathsf{T}\mathbf{P}_j\boldsymbol{\Phi}_j + \alpha\mathbf{I})^{-1}\mathbf{y}_j.$$

Therefore, taking account for the regularization in Proposition 3, for any epoch $j$,

$$\sum_{t=t_j}^{t_{j+1}-1} \left(\boldsymbol{\phi}_t^\mathsf{T}\overline{\mathbf{w}}_j - y_t\right)^2 = -\alpha\|\overline{\mathbf{w}}_j\|_2^2 + \sum_{t=t_j}^{t_{j+1}-1} \left(\boldsymbol{\phi}_t^\mathsf{T}\overline{\mathbf{w}}_j - y_t\right)^2 + \alpha\|\overline{\mathbf{w}}_j\|_2^2$$

$$= -\alpha\|\overline{\mathbf{w}}_j\|_2^2 + \alpha\mathbf{y}_j^\mathsf{T}(\boldsymbol{\Phi}_j^\mathsf{T}\mathbf{P}_j\boldsymbol{\Phi}_j + \alpha\mathbf{I})^{-1}\mathbf{y}_j.$$

Now using the kernel approximation guarantees of Proposition 1 we have

$$
\alpha \mathbf{y}_j^\mathsf{T} (\boldsymbol{\Phi}_j^\mathsf{T} \mathbf{P}_j \boldsymbol{\Phi}_j + \alpha \mathbf{I})^{-1} \mathbf{y}_j \le \alpha \mathbf{y}_j^\mathsf{T} \left( \boldsymbol{\Phi}_j^\mathsf{T} \boldsymbol{\Phi}_j - \frac{\gamma \varepsilon}{1-\varepsilon} \mathbf{I} + \alpha \mathbf{I} \right)^{-1} \mathbf{y}_j
$$

$$
= \left( \alpha - \frac{\varepsilon}{1-\varepsilon} \gamma \right)^{-1} \alpha \left( \alpha - \frac{\varepsilon}{1-\varepsilon} \gamma \right) \mathbf{y}_j^\mathsf{T} \left( \boldsymbol{\Phi}_j^\mathsf{T} \boldsymbol{\Phi}_j + \left( \alpha - \frac{\varepsilon}{1-\varepsilon} \gamma \right) \mathbf{I} \right)^{-1} \mathbf{y}_j
$$

$$
= \left( \left( \alpha - \frac{\varepsilon}{1-\varepsilon} \gamma \right)^{-1} \alpha \right) \alpha' \mathbf{y}_j^\mathsf{T} \left( \boldsymbol{\Phi}_j^\mathsf{T} \boldsymbol{\Phi}_j + \alpha' \mathbf{I} \right)^{-1} \mathbf{y}_j
$$

$$
= (1 + \varepsilon') \alpha' \mathbf{y}_j^\mathsf{T} \left( \boldsymbol{\Phi}_j^\mathsf{T} \boldsymbol{\Phi}_j + \alpha' \mathbf{I} \right)^{-1} \mathbf{y}_j,
$$

where we denoted $\varepsilon' = \left( \left( \alpha - \frac{\gamma \varepsilon}{1-\varepsilon} \right)^{-1} \alpha \right) - 1$ and $\alpha' = \left( \alpha - \frac{\gamma \varepsilon}{1-\varepsilon} \right)$. Putting it together,

$$
\sum_{t=t_j}^{t_{j+1}-1} \left( \boldsymbol{\phi}_t^\mathsf{T} \overline{\mathbf{w}}_j - y_t \right)^2 \le -\alpha \|\overline{\mathbf{w}}_j\|_2^2 + (1 + \varepsilon') \alpha' \mathbf{y}_j^\mathsf{T} \left( \boldsymbol{\Phi}_j^\mathsf{T} \boldsymbol{\Phi}_j + \alpha' \mathbf{I} \right)^{-1} \mathbf{y}_j
$$

$$
= -\alpha \|\overline{\mathbf{w}}_j\|_2^2 + (1 + \varepsilon') \left( \sum_{t=t_j}^{t_{j+1}-1} \min_{\mathbf{w} \in \mathcal{H}} \left( \boldsymbol{\phi}_t^\mathsf{T} \mathbf{w}_j - y_t \right)^2 + \alpha' \|\mathbf{w}_j\|_2^2 \right)
$$

$$
\le -\alpha \|\overline{\mathbf{w}}_j\|_2^2 + (1 + \varepsilon') \left( \sum_{t=t_j}^{t_{j+1}-1} \min_{\mathbf{w} \in \mathcal{H}} \left( \boldsymbol{\phi}_t^\mathsf{T} \mathbf{w}_j - y_t \right)^2 + \alpha \|\mathbf{w}_j\|_2^2 \right)
$$

$$
= -\alpha \|\overline{\mathbf{w}}_j\|_2^2 + (1 + \varepsilon') \left( \sum_{t=t_j}^{t_{j+1}-1} \left( \boldsymbol{\phi}_t^\mathsf{T} \widehat{\mathbf{w}}_j - y_t \right)^2 + \alpha \|\widehat{\mathbf{w}}_j\|_2^2 \right)
$$

$$
= -\alpha \|\overline{\mathbf{w}}_j\|_2^2 + \sum_{t=t_j}^{t_{j+1}-1} \left( \boldsymbol{\phi}_t^\mathsf{T} \widehat{\mathbf{w}}_j - y_t \right)^2 + \varepsilon' \alpha \|\widehat{\mathbf{w}}_j\|_2^2 + \varepsilon' \left( \sum_{t=t_j}^{t_{j+1}-1} \left( \boldsymbol{\phi}_t^\mathsf{T} \widehat{\mathbf{w}}_j - y_t \right)^2 + \alpha \|\widehat{\mathbf{w}}_j\|_2^2 \right).
$$

Therefore, extracting the $B_j$ part of the regret we get

$$
B_j \le -\alpha \|\overline{\mathbf{w}}_j\|_2^2 + \varepsilon' \alpha \|\widehat{\mathbf{w}}_j\|_2^2 + \varepsilon' \left( \sum_{t=t_j}^{t_{j+1}-1} \left( \boldsymbol{\phi}_t^\mathsf{T} \widehat{\mathbf{w}}_j - y_t \right)^2 + \alpha \|\widehat{\mathbf{w}}_j\|_2^2 \right).
$$

**Bounding $C_j$**  Changing slightly the regularizers that we add and subtract in the bound on $C_j$ we obtain

$$
\sum_{j=1}^{J} B_j + C_j = \sum_{j=1}^{J} -\alpha \|\overline{\mathbf{w}}_j\|_2^2 + \varepsilon' \alpha \|\mathbf{w}\|_2^2 + \varepsilon' \mathcal{L}_j^*.
$$

Integrating this with the bound for $A_j$ obtained in the proof of Thm 1 we get

$$
\sum_{t=1}^{T} \ell_t \left( \widetilde{\boldsymbol{\phi}}_t^\mathsf{T} \widetilde{\boldsymbol{\omega}}_t \right) - \ell_t \left( \boldsymbol{\phi}_t^\mathsf{T} \mathbf{w} \right) \le \left( \sum_{j=1}^{J} \frac{4}{\sigma_{\min}} d_{\mathrm{eff}}^j \left( \frac{\alpha}{\sigma_{\min} L^2} \right) \log \left( 2 \sigma_{\min} L^2 \operatorname{Tr}(\mathbf{K}_j)/\alpha \right) + \varepsilon' \mathcal{L}_j^* \right)
$$

$$
+ JLC + \frac{JL^2}{\alpha} + J \varepsilon' \alpha \|\mathbf{w}\|_2^2.
$$