[Reviews · NeurIPS 2017]

Reviewer 1



The paper proposes an efficient second-order online kernel learning mainly by combining KONS and Nystrom method. NOVELTY The novelty is limited on both the methodological and theoretical contributions. The achieved results do not have profound implication for the advancement of theory and practice. WRITING QUALITY The English writing and organization of this paper are relatively good. The reviewer strongly suggests the authors arrange Table 2 in the main paper rather than in Appendix because the experimental results in Table 2 are the core material. COMMENTS This paper works on accelerating second-order online kernel learning and matches the desired state-of-the-art regret bound. The proposed method improves KNOS by using some efficient Nystrom methods instead of the true kernel matrix. However, it seems that this manuscript has not been completed, and many time and accuracy comparisons in Table 1 have not been reported. Although this paper claims that the effective dimension is too large to complete the results in line 341, this explanation for the incomplete experiments is not convincing. If the effective dimension is the concern, this paper should provide more theoretical and practical analysis to show how it affects the PROS-N-KONS. In addition, the reviewer does not find the complexity comparisons among the compared methods. Thus, it is not convincing at all t that the proposed method is efficient. Also, the reviewer has three concerns on this paper: 1) In the experimental setup, how to determine the values of beta, epsilon, and bandwidth in the kernel matrix? For beta and epsilon, the related parameter selection will take much time. If the learning performance is very sensitive to these two parameters, the efficiency of the proposed method decreases. Regarding the bandwidth, it has an effect on both the effective dimension and the learning accuracy. 2) The experimental comparison is not convincing. The efficiency is the key contribution of the proposed methods, but the run time comparison with baselines is missing. Moreover, the Dual-SGD is conducted in only one selected dataset, which makes the model comparison less reasonable. 3) For the ridge leverage scores, it is surprising to find that the paper does not cite paper [Alaoui], which originally proposes the ridge leverage scores in Nystrom approximation with the application to regression analysis. Actually, to my understanding, [Alaoui] only shows the size in Nystrom approximation could be reduced while does not claim their method would be faster than previous methods, and therein the leverage score approximation in Theorem 4 could cost much time even more than the time spent in Theorem 3 (just check how to satisfy the condition of his Theorem 3 and 4). Hope this concern could be helpful to tackle your negative claim and results in line 341. [Alaoui] El Alaoui, A., Mahoney, M. W. (2014). Fast randomized kernel methods with statistical guarantees. Advances in Neural Information Processing Systems.

Reviewer 2



In the paper, the authors propose a second-order online kernel learning that achieves logarithm regret under directional curvature assumption. This algorithm can be viewed as an extension of previous two works: Efficient second order online learning by sketching [12] and Second-order kernel OCO with adaptive sketching [1] (especially the latter one), while the major innovation of this paper is replacing the high-dimensional feature map $\phi$ with a finite-dimensional approximation $\bar{\phi}$. Most of the notations are inherited from [1], like RLS and effective dimension, and the authors also give a fair comparison to the one in [1] in terms of the theoretical regret. However, this cannot be seen from the experimental parts since the algorithm in [1] is not compared as a competitor. The whole comparison is rather arbitrary since this algorithm is majorly compared with FOGD and NOGD, while Dual-SGD is occasionally compared. Besides, the second-order methods are generally slower than first-order methods, but the authors do not report the calculation time for FOGD and NOGD. Besides, a minor error in Line 114.

Reviewer 3



The authors study the problem of online kernel learning for the purposes of improved efficiency. They remark that first-order methods accumulate O(\sqrt{T}) regret with a per-step cost of O(t) while second-order methods accumulative O(\log(T)) regret with a per-step cost of O(t^2). To improve upon this, the authors propose to perform a Nystrom sketch of the feature map and then perform efficient second-order updates on this approximate RKHS. The authors show that the size of the embedding depends on the effective dimension of the data, which can be much smaller than t. The paper also includes experiments on real datasets showing that their method and some heuristic variants of it are competitive with both first and second-order methods on large data. This paper is overall well-written and clear. The authors do a good job of motivating the problem they are trying to solve, explaining why their solution is intuitive, and then explaining the theoretical guarantees and how they compare to existing results. The theoretical result is interesting and compelling, and the experimental results seem to validate the theory. A few questions and comments: 1) Lines 173-175: The authors assert that using approximate updates and a fixed budget of SV would lose the log(T) regime because the reduction in step-size would slow down. Can they explain this in more detail or provide a reference to this? 2) Lines 285-286: The authors claim that better bounds can be obtained by optimizing Eqn 3 over \gamma. However, isn't d_eff^T(\gamma) unknown? That would make this ideal impossible, especially in an online setting. 3) Table 1: On the slice dataset, doesn't FOGD outperform B-KONS? 4) Lines 387-349: The authors argue that frequent restarts prevent PROS-N-KONS from performing well on the smaller datasets. However, doesn't CON-KONS avoid restarts, while performing similarly to PROS-N-KONS? 5) Citation [7] and [8] appear to be the same.